# Comparative Study of Three Steganographic Methods Using a Chaotic System and Their Universal Steganalysis Based on Three Feature Vectors

**DOI:** 10.3390/e21080748

**Published:** 2019-07-30

**Authors:** Dalia Battikh, Safwan El Assad, Thang Manh Hoang, Bassem Bakhache, Olivier Deforges, Mohamad Khalil

**Affiliations:** 1LASTRE Laboratory, Lebanese University, 210 Tripoli, Lebanon; 2Institut d’Electronique et des Télécommunications de Rennes (IETR), UMR CNRS 6164, Université de Nantes—Polytech Nantes, Rue Christian Pauc CS 50609, CEDEX 3, 44306 Nantes, France; 3School of Electronics and Telecommunications, Hanoi University of Science and Technology, 1 Dai Co Viet, Hai Ba Trung, Hanoi, Vietnam; 4INSA de Rennes, CNRS, IETR, CEDEX 7, 35708 Rennes, France

**Keywords:** steganography, chaotic system, steganalysis, wavelet, feature vector, SVM, FLD

## Abstract

In this paper, we firstly study the security enhancement of three steganographic methods by using a proposed chaotic system. The first method, namely the Enhanced Edge Adaptive Image Steganography Based on LSB Matching Revisited (EEALSBMR), is present in the spatial domain. The two other methods, the Enhanced Discrete Cosine Transform (EDCT) and Enhanced Discrete Wavelet transform (EDWT), are present in the frequency domain. The chaotic system is extremely robust and consists of a strong chaotic generator and a 2-D Cat map. Its main role is to secure the content of a message in case a message is detected. Secondly, three blind steganalysis methods, based on multi-resolution wavelet decomposition, are used to detect whether an embedded message is hidden in the tested image (stego image) or not (cover image). The steganalysis approach is based on the hypothesis that message-embedding schemes leave statistical evidence or structure in images that can be exploited for detection. The simulation results show that the Support Vector Machine (SVM) classifier and the Fisher Linear Discriminant (FLD) cannot distinguish between cover and stego images if the message size is smaller than 20% in the EEALSBMR steganographic method and if the message size is smaller than 15% in the EDCT steganographic method. However, SVM and FLD can distinguish between cover and stego images with reasonable accuracy in the EDWT steganographic method, irrespective of the message size.

## 1. Introduction

Steganography is an increasingly important security domain; it aims to hide a message (secret information) in digital cover media without causing perceptual degradation (in this study, we use images as cover media). It should be noted that many steganographic methods have been proposed in the spatial and frequency domains. In the spatial domain, pixels are directly used to hide secret messages; these techniques are normally easy to implement and have a high capacity. However, they are not generally robust against statistical attacks [1,2]. In the transform domain, coefficients of frequency transforms, such as DCT (Discrete Cosine Transform), FFT (Fast Fourier Transform), and DWT (Discrete Wavelet Transform), are used to hide secret data. Generally, these techniques are complex, but they are more robust against steganalysis (to noise and to image processing).

The main steganographic methods in the spatial domain [3,4,5,6,7,8,9,10,11,12,13,14,15,16,17] are LSB-based (Low Significant Bit). Recently, entropy has also been extensively used to support data-hiding algorithms [18,19,20]. The LSB methods entail replacing the least significant bit of pixels with a bit of the secret data. Among these methods, the EALSBMR method [3] is an edge adaptive scheme with respect to the message size and can embed data according to the difference between two consecutive pixels in the cover image. To the best of our knowledge, we conclude that this method is the best (good PNSR, high embedding capacity, and especially adaptive), but it suffers from low security in terms of message detection. For this reason, we have enhanced its security.

Frequency domain steganography, as a watermarking domain [21,22,23,24,25,26,27,28,29], is widely based on the DCT and DWT transforms. The DCT usually transforms an image representation into a frequency representation by grouping pixels into 8 × 8 pixel blocks and transforming each block, using the DCT transform, into 64 DCT coefficients. A message is then embedded into the DCT coefficients. The Forward Discrete Wavelet Transform is, in general, suitable for identifying areas in the cover image where a secret message can be effectively embedded due to excellent space-frequency localization properties. In particular, these properties allow exploiting the masking effect of a human visual system so that if a DWT coefficient is modified, it modifies only the region that corresponds to that coefficient. The Haar wavelet is the simplest possible wavelet that can achieve the DWT.

However, the aforementioned steganographic methods are not secure in terms of message detection. To protect the content of messages, chaos can be used. Indeed, chaotic sequences play an important role in information hiding and in security domains, such as cryptography, steganography, and watermarking, because of their properties such as sensitivity to initial conditions and parameters of the system, ergodicity, uniformity, and pseudo-randomness. Steganography generally leaves traces that can be detected in stego images. This can allow an adversary, using steganalysis techniques, to divulge a hiding secret message. There are two types of opponents: passive and active. A passive adversary only examines communication to detect whether communication contains hidden messages. In this case, the content of the communication is not modified by the rival. An active adversary can intentionally cause disruption, distortion, or destruction of communication, even in the absence of evidence of secret communication. The main steganographic methods have been designed for cases of passive adversary. In general, there are two kinds of steganalysis: specific and universal. Specific steganalysis is designed to attack a specific steganography algorithm. This type of specific steganalysis can generally produce more accurate results, but it fails to produce satisfactory results if the inserted secret messages are in the form of a modified algorithm. Universal steganalysis, on the other hand, can be regarded as a universal technique to detect various types of steganography. Moreover, it can be used to detect new steganographic techniques where specific steganalysis does not yet exist. In other words, universal steganalysis is an irreplaceable tool for detection if the integration algorithm is unknown or secret.

In this paper, we first integrate an efficient chaotic system into the three steganographic methods mentioned above to make them more secure. The chaotic system quasi-chaotically chooses pixel positions in the cover image where the bits of the secret message will be embedded. Thus, the inserted bits of the secret message becomes secure against message bits recovery attacks because their position is unknown.

Second, we study and apply three universal steganalysis methods to the aforementioned chaos-based steganographic methods. The first steganalysis method, developed by Farid [30], uses higher-order statistics of high-frequency wavelet sub-bands and their prediction errors to form the feature vectors. In the second steganalysis method, as formulated by Shi et al. [31], the statistical moments of the characteristic functions of the prediction-error image, the test image, and their wavelet sub-bands are selected as the feature vectors. The third steganalysis method, introduced by Wang et al. [32], uses the features that are extracted from both the empirical probability density function (PDF) moments and the normalized absolute characteristic function (CF). For the three steganalysis algorithms, we applied FLD analysis and the SVM method with the RBF kernel as classifiers between cover images and stego images.

The paper has been organized as follows: In Section 2, we describe the proposed chaotic system. In Section 3, we present the three enhanced steganographic algorithms. In Section 4, we illustrate the experimental results and analyze the enhanced algorithms. In Section 5, we develop, in detail, the steganalysis techniques for the previous algorithms. In Section 6, we report the results of the steganalysis, and in the last section, we conclude our work.

## 2. Description of the Proposed Chaotic System

This system is made of a perturbed chaotic generator and a 2-D cat map. The chaotic generator supplies the dynamic keys $K_{p}$ for the process of provides the position of the new random pixel (see Figure 1). The chaotic system allows inserting a message both in a secretive and uniform manner [33,34,35,36,37,38,39,40].

The generator of discrete chaotic sequences exhibits orbits with very large lengths. It is based on two connected non-linear digital IIR filters (cells). The discrete PWLCM and SKEW TENT maps (non-linear functions) are used. A linear feedback shift register (m-LFSR) is then used to disturb each cell (Figure 2). The disturbing technique is associated with the cascading technique, which allows controlling and increasing the length of the orbits that are produced. The minimum orbit length of the generator output is calculated using Equation (Equation 1):(1)$$\;o_{min}=lcm\left\{\Delta_{1}\times \left(2^{k_{1}}-1\right),\Delta_{2}\times \left(2^{k_{2}}-1\right)\right\}$$

In the above equation, lcm is the least common multiple, $k_{1}$ = 23 and $k_{2}$ = 21 are the degrees of the LFSR’s primitive polynomials, and $\Delta_{1}$ and $\Delta_{2}$ are the lengths $s_{1}$ and $s_{2}$ of outputs cells, respectively, without disturbance. The equations of the chaotic generators are formulated as follows:(2)$$\begin{array}{c} s_{i}\left(n\right)=NLF_{i}\left\{u_{i}\left(n-1\right),p_{i}\right\},i=1,2\\ u_{i}\left(n-1\right)=mod\left\{s_{i}\left(n-1\right)\times c_{i,1}+s_{i}\left(n-2\right)\times c_{i,2},2^{N}\right\},i=1,2\\ s\left(n\right)=s_{1}\left(n\right)+s_{2}\left(n\right) \end{array}$$

The two previously mentioned functions, PWLCM map and Skew map, are defined according to the following relations:(3)$$\begin{array}{cc} s_{1}\left(n\right) & =NLF_{1}\left\{u_{1}\left(n-1\right),p_{1}\right\}\\ & =\left\{\begin{array}{c} \left\lfloor 2^{N}\times \frac{u_{1}\left(n-1\right)}{p_{1}}\right\rfloor \;if\;0\leq u_{1}\left(n-1\right)<p_{1}\\ \left\lfloor 2^{N}\times \frac{2^{N}-u_{1}\left(n-1\right)}{2^{N}-p_{1}}\right\rfloor \;if\;p_{1}\leq u_{1}\left(n-1\right)<2^{N-1}\\ NLF_{1}\left[2^{N}-u_{1}\left(n-1\right)\right]\;otherwise \end{array}\right. \end{array}$$
(4)$$\begin{array}{cc} s_{2}\left(n\right) & =NLF_{2}\left[u_{2}\left(n-2\right),p_{2}\right]\\ & =\left\{\begin{array}{c} \left\lfloor 2^{N}\times \frac{u_{2}\left(n-1\right)}{p_{1}}\right\rfloor \;if\;0\leq u_{2}\left(n-1\right)<p_{2}\\ \left\lfloor 2^{N}\times \frac{2^{N}-u_{2}\left(n-1\right)}{2^{N}-p_{2}}\right\rfloor +1\;if\;p_{2}\leq u_{2}\left(n-1\right)<2^{N} \end{array}\right. \end{array}$$

The control parameter $p_{1}$ is used for the PWLCM map and ranges from 1 to $2^{N-1}-1$, and $p_{2}$ is the control parameter that is used for the Skew map and ranges from 1 to $2^{N}-1$. N=32 is the word length used for simulations. The size of the secret key *K*, formed by all initial conditions and parameters of the chaotic generator, is (6 × 32 + 5 × 32 + 31 + 23 +21) = 427 bits. It is large enough to resist a brute-force attack.

### Description of the Cat Map Used

The permutation process is based on the modified Cat map and is calculated in a very efficient manner using the equation below [37]: (5)$$\left[\begin{array}{c} M_{cn}\\ M_{ln} \end{array}\right]=mod\left\{\left(\begin{array}{cc} 1 & u\\ v & 1+uv \end{array}\right)\times (\begin{array}{c} M_{l}\\ M_{c} \end{array})+\left[\begin{array}{c} r_{l}+r_{c}\\ r_{c} \end{array}\right],[\begin{array}{c} M\\ M \end{array}]\right\}+\left[\begin{array}{c} 1\\ 1 \end{array}\right]$$

In the above equation, $\left(M_{l},M_{c}\right)$ and $\left(M_{ln},M_{cn}\right)$ are the original and permuted square matrices of size $\left(M,M\right)$, from which we calculate the Ind matrix as follows:$$M_{l}=\left(\begin{array}{ccccc} 1 & 1 & . & .\; & 1\\ 2 & 2 & . & . & 2\\ . & & . & & .\\ . & & . & & .\\ M & M & . & . & M \end{array}\right);M_{c}=\left(\begin{array}{ccccc} 1 & 2 & . & .\; & M\\ 1 & 2 & . & . & M\\ . & & . & & .\\ . & & . & & .\\ 1 & 2 & . & . & M \end{array}\right)$$
$$Ind=\left(M_{ln}-1\right)+\left(M_{cn}-1\right)\times M+1$$

The dynamic key $K_{p}$ is structured as follows:$$K_{p}=\left[k_{p_{1}},k_{p_{2}},\ldots ,k_{p_{r}}\right]$$
$$k_{p_{i}}=\left\{u_{i},v_{i},rl_{i},rc_{i}\right\};i=1,2,\ldots ,r$$

In the above equations, $0\leq u_{i},v_{i},rl_{i},rc_{i}\leq M-1$ are the parameters of the Cat map and *r* is the number of rounds.

## 3. Enhanced Steganographic Algorithms

In this section, we describe three enhanced steganographic algorithms by using an efficient chaotic system.

### 3.1. Enhanced EALSBMR (EEALSBMR)

Below, we present the insertion procedure and the extraction procedure of the proposed enhancement of the EALSBMR method (EEALSBMR) [41].

#### 3.1.1. Insertion Procedure

The flow diagram of the embedding scheme can be found in Figure 3.

The detailed embedding steps for this algorithm have been explained as follows:Step 1:Capacity estimation
To estimate the insertion capacity, we arrange the cover image into a 1D vector *V*, and we divide its content into non-overlapping embedding units (blocks) consisting of two consecutive pixels $\left(p_{i},p_{i+1}\right)$. Following this, we calculate the difference between the pixels of each block, and we increase by one the content of the vector-difference VD of 31 elements $t\in \left\{1,2,3,\ldots ,31\right\}$, in which each element contains $\left|EU\left(t\right)\right|$ number of blocks where $EU\left(t\right)$ is a set of pixel pairs whose absolute differences are greater than or equal to *t*, as shown below:
(6)$$EU\left(t\right)=\left\{\left(p_{i},p_{i+1}\right)\left|\right|\left|p_{i}-p_{i+1}\right|\geq t,\forall \left(p_{i},p_{i+1}\right)\in V\right\}$$For a given secret message *M* of size $\left|M\right|$ bits, the threshold *T* used in the embedding process is determined by the following expression and pseudo-code (Algorithm 1):
(7)$$T=argmax_{t}\left\{2*\left|EU\left(t\right)\right|\geq \left|M\right|\right\}$$
**Algorithm 1** Pseudo-code determining the value of the threshold *T*  1:**procedure**  2:    number_pixels = 0;  3:    **for**
*t* = 31:-1:1 **do**  4:        number_pixels = number_pixels + VD(t);  5:        **if** (2*number_pixels > = |M|) **then**  6:           T=t;  7:           break;  8:        **end if**;  9:    **end for**;10:**end procedure**Step 2:Embedding process
The embedding process is achieved as follows: we divide the cover image into two sub-images; one includes the odd columns, and the other includes the even columns.Following this, the chaotic system chooses a pixel position (Ind) from the odd sub-image; the second pixel position of the corresponding block must have the same Ind in the even image. If a pair of pixel units $\left(p_{i},p_{i+1}\right)$ satisfies Equation (Equation 8), then a 2 bit-message can be hidden (one bit by pixel); otherwise, the chaotic system chooses another Ind.
(8)$$\left(\left|p_{i}-p_{i+1}\right|\geq T,\forall \left(p_{i},p_{i+1}\right)\in V\right)$$For each unit $\left(p_{i},p_{i+1}\right)$, we perform data-hiding based on the following four cases [42]:
Case 1:if $LSB\left(p_{i}\right)=m_{i}$ and $f\left(p_{i},p_{i+1}\right)=m_{i+1}$$\to \left(p_{i}^{{}^{\prime }},p_{i+1}^{{}^{\prime }}\right)=\left(p_{i},p_{i+1}\right)$Case 2:if $LSB\left(p_{i}\right)=m_{i}$ and $f\left(p_{i},p_{i+1}\right)\neq m_{i+1}$$\to \left(p_{i}^{{}^{\prime }},p_{i+1}^{{}^{\prime }}\right)=\left(p_{i},p_{i+1}+r\right)$Case 3:if $LSB\left(p_{i}\right)\neq m_{i}$ and $f\left(p_{i}-1,p_{i+1}\right)=m_{i+1}$$\to \left(p_{i}^{{}^{\prime }},p_{i+1}^{{}^{\prime }}\right)=\left(p_{i}-1,p_{i+1}\right)$Case 4:if $LSB\left(p_{i}\right)\neq m_{i}$ and $f\left(p_{i}-1,p_{i+1}\right)\neq m_{i+1}$$\to \left(p_{i}^{{}^{\prime }},p_{i+1}^{{}^{\prime }}\right)=\left(p_{i}+1,p_{i+1}\right)$In the above equations, $m_{i}$ and $m_{i+1}$ are the $i^{}$th and ${\left(i+1\right)}^{}$th secret bits of the message to be embedded; *r* is a random value belonging to $\left\{-1,1\right\}$, and $\left(p_{i}^{{}^{\prime }},p_{i+1}^{{}^{\prime }}\right)$ denotes the pixel pair after data-hiding. The function *f* is defined as follows:
(9)$$f\left(a,b\right)=LSB\left(\left\lfloor \frac{a}{2}\right\rfloor +b\right)$$Readjustment if necessary: After hiding, $\left(p_{i}^{{}^{\prime }},p_{i+1}^{{}^{\prime }}\right)$ may be out of range [0, 255] or the new difference value $\left|p_{i}^{{}^{\prime }}-p_{i+1}^{{}^{\prime }}\right|$ may be less than the threshold *T*. In these cases, we need to readjust $p_{i}^{{}^{\prime }}$ and $p_{i+1}^{{}^{\prime }}$, and the new readjusted values, $p_{i}^{\prime \prime }$ and $p_{i+1}^{\prime \prime }$, are calculated as follows [3]:
(10)$$\left(\;p_{i}^{\prime \prime },p_{i+1}^{\prime \prime }\right)=argmin_{\left(e_{1},e_{2}\right)}\left\{\left|e_{1}-p_{i}^{{}^{\prime }}\right|+\left|e_{2}-p_{i+1}^{{}^{\prime }}\right|\right\}$$
with :
(11)$$\left\{\begin{array}{c} e_{1}=p_{i}^{{}^{\prime }}+4k_{1}\\ e_{2}=p_{i+1}^{{}^{\prime }}+2k_{2} \end{array}\right.k_{1},k_{2}\in Z$$
$k_{1},k_{2}$ are two arbitrary numbers from Z; when:
(12)$$0\leq e_{1},e_{1}\leq 255\;\mathrm{and}\;\left|e_{1}-e_{2}\right|\geq T$$
then :
(13)$$\begin{array}{c} p_{i}^{{}^{\prime \prime }}=e_{1}\\ \;p_{i+1}^{{}^{\prime \prime }}=e_{2} \end{array}$$The sequence follows as such for each new block position.Finally, we embed the parameter *T* of the stego image into the first five pixels or the last five pixels, for example.

#### 3.1.2. Extraction Procedure

Extract the parameter *T* from the stego image.Divide the stego image into two sub-images; one includes the odd columns, and the other includes the even columns.Generate a pseudo-chaotic position (using the same secret key *K*), as done in the insertion procedure, to obtain the same order of pixel unit position as the odd sub-image. The second pixel block has the same Ind in the even image.Verify if $\left|p_{i}^{s}-p_{i+1}^{s}\right|\geq T$ and then extract the two secret bits of *M*$\left(m_{i},m_{i+1}\right)$ as follows:
(14)$$m_{i}=LSB\left(p_{i}^{s}\right);\;m_{i+1}=f\left(p_{i}^{s},p_{i+1}^{s}\right)$$
with : $p_{i}^{s}=p_{i}^{{}^{\prime }}\;$ or $\;p_{i}^{\prime \prime }$Otherwise, the chaotic system chooses another pseudo-chaotic position. The sequence follows as such for each unit position until all messages have been extracted.Example of insertion:The cover image is this image of “peppers” as in Figure 4:The embedded message appears as follows in 40 × 40 pixels as shown in Figure 5:The corresponding sequence of the bits message has been given as follows:
M=10001000100011001000110001100111001001111010010110
11101011000110101011101000000110100010110010…The length of the binary message is 13,120 bits.Capacity estimation produces the threshold T=12Suppose that the pseudo-chaotic positions of a block to embed the two bits message $m_{1}=1$ and $m_{2}=0$ are (354, 375) and (354, 376) that correspond to the 141 and 129 gray values (see Figure 6).Hiding the message bits:
$$LSB\left(141\right)=1=m_{1}=1$$
$$f\left(p_{1},p_{2}\right)=LSB\left(\left\lfloor \frac{p_{1}}{2}\right\rfloor +p_{2}\right)=LSB\left(70+129\right)=1\neq m_{2}$$We are in Case 2:
$$LSB\left(p_{i}\right)=m_{i};f\left(p_{i},p_{i+1}\right)\neq m_{i+1}$$Therefore, the new pixel values are as follows:
$$\left(p_{1}^{{}^{\prime }},p_{2}^{{}^{\prime }}\right)=\left(p_{1},p_{2}+r\right)=\left(141,130\right)\;\mathrm{with}\;r=1$$The difference between the new pixel values is:
$$d^{{}^{\prime }}=\left|p_{1}^{{}^{\prime }}-p_{2}^{{}^{\prime }}\right|=\left|141-130\right|=11<T$$Then we need to adjust the new pixel values:We test the values $-50<k_{1}<50$ and $-50<k_{2}<50$ until we obtain the smallest difference between the initial values $p_{1}^{{}^{\prime }}$ and $p_{2}^{{}^{\prime }}$ and the corresponding obtained values $e_{1}$ and $e_{2}$ by using Equations (Equation 12) and (Equation 13). In our example, we find $k_{1}=0$ and $k_{2}=-1$ and then: $p_{1}^{\prime \prime }=141$, $p_{2}^{\prime \prime }=128$.Extraction of the bits message in the previous insertion example:The extraction is performed using the following equation:
$$m_{1}=LSB\left(p_{1}^{\prime \prime }\right)=LSB\left(141\right)=1$$
$$m_{2}=f\left(p_{1}^{\prime \prime },p_{2}^{\prime \prime }\right)=LSB\left(\left\lfloor \frac{p_{1}^{\prime \prime }}{2}\right\rfloor +p_{2}^{\prime \prime }\right))=LSB\left(70+128\right)=LSB\left(198\right)=0$$

### 3.2. Enhanced DCT Steganographic Method (EDCT)

The DCT transforms a signal or image from the spatial domain into the frequency domain [43,44]. A DCT expresses a sequence of finitely many data points in terms of a sum of cosine functions, oscillating at different frequencies. The 2D DCT is calculated as follows:(15)$$DCT_{i,j}=\alpha_{i}\alpha_{j}\sum_{m=0}^{M-1}\sum_{n=0}^{N-1}C_{mn}\cos\frac{\pi (2m+1)i}{2M}\cos\frac{\pi (2n+1)j}{2N}$$
where:$$\alpha_{i}=\left\{\begin{array}{c} \frac{1}{\sqrt{M}}\;i=0\\ \sqrt{\frac{2}{M}}\;0\leq i\leq \;M-1 \end{array}\right.\;\alpha_{j}=\left\{\begin{array}{c} \frac{1}{\sqrt{N}}\;i=0\\ \sqrt{\frac{2}{N}}\;0\leq i\leq \;N-1 \end{array}\right.$$

The block diagram of the proposed enhanced steganographic-based DCT transform has been shown in Figure 7.

#### 3.2.1. Insertion Procedure

The embedding process consists of the following steps:Read the cover image and the secret message.Convert the secret message into a 1-D binary vector.Divide the cover image into 8 × 8 blocks. Then apply the 2D DCT transformation to each block (from left to right, top to bottom).Use the same chaotic system to generate a pseudo-chaotic Ind.Replace the LSB of each located DCT coefficient with the one bit of the secret message to hide.Apply the 2D Inverse DCT transform to produce the stego image.

#### 3.2.2. Extraction Procedure

The extraction procedure consists of the following steps:Read the stego image.Divide the stego image into 8 × 8 blocks and then apply the 2D DCT to each block.Use the same chaotic system to generate pseudo-chaotic Ind.Extract the LSB of each pseudo-located coefficient.Construct the secret image.

### 3.3. Enhanced DWT Steganographic Method (EDWT)

The embedded secret image in the lower frequency sub-band (A) is generally more robust than the other sub-bands, but it significantly decreases the visual quality of the image, as normally, most of the image energy is stored in this sub-band. In contrast, the edges and textures of the image and the human eye are not generally sensitive to changes in the high-frequency sub-band (D); this allows secret information to be embedded without being perceived by the human eye. However, the sub-band (D) is not robust against active attacks (filtering, compression, etc.). The compromise adopted by many DWT-based algorithms to achieve accepted performance of imperceptibility and robustness enables embedding the secret image in the middle-frequency sub-bands (H) or (V). In the block diagram of the proposed steganographic EDWT method shown in Figure 8, we embed the secret image in the sub-band (H) of the cover image (the size of the secret message must, at most, be equal to the size of the sub-band (H) of the cover image).

#### 3.3.1. Insertion Procedure

The embedding process consists of the following steps:Read the cover image and the secret image.Transform the cover image into one level of decomposition using Haar Wavelet.Permute the secret image in a pseudo-chaotic manner.Fuse the DWT coefficients (H) of the cover image and the permuted secret image PSI as follows [45]:
(16)$$\begin{array}{c} X^{{}^{\prime }}=\alpha X+\beta \times PSI\\ \alpha +\beta =1;\;\alpha \gg \beta \end{array}$$In the above equations, $X^{{}^{\prime }}$ is the modified DWT coefficient (H); *X* is the original DWT coefficient (H). α and β are the embedding strength factors; they are chosen such that the resulting stego image has a large PSNR. In our experiments, we tested some values of β, and the best value was found to be approximately 0.01.Apply Inverse Discrete Wavelet Transform (IDWT) to produce the stego image in the spatial domain.

#### 3.3.2. Extraction Procedure

The extraction procedure involves the following steps:Read the stego image.Transform the stego image into one level of decomposition using Haar Wavelet.Apply inverse fusion transform to extract the permuted secret image as follows:
(17)$$PSI=(X^{{}^{\prime }}-\alpha X)/\beta$$The extraction procedure is not blind, as we need the cover image to extract the permuted secret message.Apply the inverse permutation procedure using the same chaotic system to obtain the secret image.

## 4. Experimental Results and Analysis

In the experiments, we first create the stego images by using the implemented steganographic methods that were applied on the standard gray level cover images “Lena”, “Peppers”, “Baboon” in 512 × 512 pixels and using “Boat” as a secret message with different sizes (embedding rates, ranging from 5% to 40%). The six criteria used to evaluate the qualities of the stego images have been listed as follows: Peak Signal-to-Noise Ratio (PSNR) [46], Image Fidelity (IF), structural similarity (SSIM), the entropy (*E*), the redundancy (*R*), and the image redundancy (IR). They can be represented by the following equations:(18)$$PSNR=10\times \log_{10}\left(\frac{Max\;p_{c}^{2}\left(i,j\right)}{\frac{1}{M\times N}\left(\sum_{i=0}^{M-1}\sum_{j=0}^{N-1}{\left[p_{c}\left(i,j\right)-p_{s}\left(i,j\right)\right]}^{2}\right)}\right)$$
(19)$$IF=1-\frac{\sum_{i=0}^{M-1}\sum_{j=0}^{N-1}{\left[p_{c}\left(i,j\right)\right]}^{2})}{\left(\sum_{i=0}^{M-1}\sum_{j=0}^{N-1}{\left[p_{c}\left(i,j\right)-p_{s}\left(i,j\right)\right]}^{2}\right)}$$
(20)$$SSIM=\frac{\left(2\mu_{c}\mu_{s}-1\right)\left(2cov_{cs}+c_{2}\right)}{\left(\mu_{c}^{2}+\mu_{s}^{2}+c_{1}\right)\left(\sigma_{c}^{2}+\sigma_{s}^{2}+c_{2}\right)}$$

In the above equations, $p_{c}\left(i,j\right)$ and $p_{s}\left(i,j\right)$ are the pixel value of the $i^{}$th row and $j^{}$th column of the cover and stego image; *M* and *N* are the width and height of the considered cover image.

$\mu_{c}$, $\mu_{s}$ are the average of the cover and stego images; $\sigma_{c}^{2}$, $\sigma_{s}^{2}$ are the variance of the cover and stego images; $\mu_{cs}$ is the co-variance of the cover-stego; $c_{1}={\left(k_{1}L\right)}^{2}$, $c_{2}={\left(k_{2}L\right)}^{2}$ are two variables that are used to stabilize the division with a weak denominator; *L* is the dynamic range of the pixel values, and $k_{1}$, $k_{2}$ are two much smaller constants compared to 1. We considered $k_{1}$ = $k_{2}$ = 0.05.

The higher the PSNR, IF, and SSIM, the better the quality of the stego image. PSNR values falling below 40 dB indicate a fairly low quality. Therefore, a high-quality stego should strive to be above 40 dB.

Additionally, we used three other parameters to estimate the qualities of the stego images. These parameters have been listed as follows:-The Entropy E, given by the following relation:
(21)$$E=-\sum_{0}^{2^{L}-1}p\left(P_{i}\right)log_{2}\left(p\left(P_{i}\right)\right)$$
*L* is already defined. $p(P_{i})$ is the probability of the pixel value $P_{i}$.-The Redundancy *R* is usually represented by the following formula:
(22)$$R=\frac{E_{max}-E}{E}$$
Here, $E_{max}=8$. However, this relationship is problematic because the value of the minimal entropy is not known. For that, Tasnime [47] proposed using the following relationship, which seems to be more precise:
(23)$$IR=\frac{\sum_{i=1}^{L}\left|R_{i}-R_{opt}\right|}{R_{opt}\left(2^{L}-1\right)+\left(S-R_{opt}\right)}$$Called Image Redundancy (IR) with:
*S* being the size of the image under test;$R_{i}$ being the number of occurrences of each pixel value;$R_{opt}$ being the optimal number of occurrences that each pixel value should have to get a non-redundant image.In the following section, we present and compare the performance of the three implemented steganographic methods.

### 4.1. Enhanced EALSBMR

The results obtained from the parameters PSNR, IF, and SSIM for the algorithm have been presented in Table 1; their values indicate the high quality of the stego images, even with a high embedding rate of 40%. We observe that the PSNR, IF, and SSIM values decrease, as expected, when the size of the secret message increases.

In Figure 9a–c, we show the “Baboon” cover image and the corresponding stego images for 5% and 40% embedding rates, respectively. The visual quality obtained from the “Baboon” stego images is very high because visually, it is impossible to discriminate between the cover and stego images.

Just to fix the ideas, using the Lina image as the cover, and to obtain approximately identical capacity, we globally compared the obtained PSNR of the EEALSBMP method with that obtained by the following methods: [4,5,6,17]. We observed that only the method proposed by Borislav et al. [17] produces a better PSNR than the EEALSBMP method. However, this method cannot be adapted.

### 4.2. Enhanced DCT Steganographic Method

The results obtained from this method, as presented in Table 2, indicate the high quality of the stego images, even with a high embedding rate. Additionally, even the visual quality obtained is very high, as shown in Figure 10.

### 4.3. Enhanced DWT Steganographic Method

Table 3 presents the results obtained from the EDWT algorithm, which indicate that the steganographic algorithm exhibits good performance. Furthermore, no visual trace can be found in the resulting stego images, as shown in Figure 11a–c.

### 4.4. Performance Comparison of the Three Steganographic Methods

Table 1, Table 2 and Table 3 of PSNR, IF, and SSIM of the three methods show that the EEALSBMR and EDCT methods, in comparison with the EDWT method, ensure better quality of the stego images at different embedding rates. There is approximately a 10-dB difference in PSNRs at a 5% embedding rate and a 5 to 8 dB difference in PSNRs at a 40% embedding rate.

### 4.5. Performance Using Parameters *E*, *R* and IR

The results obtained from parameters *E*, *R*, and IR for the three algorithms on the stego images with different embedding rates have been presented in Table 4, Table 5 and Table 6. As we can see, these values, given in Table 7, are too close to the values obtained over the original images. This is consistent with the previous results obtained from the parameters PSNR, IF, and SSIM regarding the high quality of the stego images.

## 5. Universal Steganalysis

A good steganographic method should be imperceptible not only to human vision systems but also to computer analysis. Steganalysis is the art and science that detects whether a given image has a message hidden in it [1,48]. The extensive range of natural images and the wide range of data embedding algorithms make steganalysis a difficult task. In this work, we consider universal steganalysis to be based on statistical analysis.

Universal (blind) steganalysis attempts to detect hidden information without any knowledge about the steganographic algorithm. The idea is to extract the features of cover images and the features of stego images and then use them as the feature vectors that are used by a supervised classifier (SVM, FLD, neural networks…) to distinguish whether the image under test is a stego image. This procedure is illustrated in Figure 12. The left side of the flowchart displays the different steps of the learning process while the right side illustrates the different steps of the testing process.

### 5.1. Multi-Resolution Wavelet Decomposition

The DWT, which uses a sub-bands coding algorithm, is found to quickly compute the Wavelet Transform. Furthermore, it is easy to implement and reduces the computation time and the number of resources required. The DWT analyses the signal at different frequency bands with different resolutions by decomposing the signal into a coarse approximation and into detailed information. The decomposition of the signal into different frequencies is achieved by applying separable low-pass $\hat{g}\left(n\right)$ and high-pass $\hat{h}\left(n\right)$ filters along the image axes. The DWT computes the approximation coefficients matrix *A* and details coefficients matrices *H*, *V*, and *D* (horizontal, vertical, and diagonal, respectively) of the input matrix *X*, as illustrated in Figure 13.

### 5.2. Feature Vector Extraction

As the amount of image data is enormous, it is not feasible to directly use the complete image data for analysis. Therefore, for steganalysis, it is useful to extract a certain amount of useful data features that represent the image instead of the image itself. The addition of a message to a cover image may not affect the visual appearance of the image, but it will affect some statistics. The features required for steganalysis should be able to detect these minor statistical disorders that are created during the data-hiding process.

Three feature-extraction techniques are used in this paper to detect the presence of a secret message; these methods calculate the statistical properties of the images by employing multi-resolution wavelet decomposition.

#### 5.2.1. Method 1: Feature Vectors Extracted from the Empirical Moments of the PDF-Based Multi-Resolution Coefficients and Their Prediction Error

The multi-resolution wavelet decomposition employed here is based on separable quadrature mirror filters (QMFs). This decomposition splits the frequency space into multiple scales and orientations. This is accomplished by applying separable low-pass and high-pass filters along the image axes, generating a vertical, horizontal, diagonal, and low-pass sub-band. The horizontal, vertical, and diagonal sub-bands at scale *m* = 1, 2, ..., *n* are denoted as $H_{m}$ , $V_{m}$ and $D_{m}$.

In our work, the first set of features is extracted from the statistics over coefficients $S_{m}$ (x,y) of each sub-band and for levels (scales) *m* = 1 and *n* = 3. These characteristics represent the following: mean μ, variance $\sigma^{2}$, skewness ξ, and kurtosis κ. They can be represented as follows:(24)$$\begin{array}{c} \mu =\frac{1}{N_{x}N_{y}}\sum_{x,y}S_{m}\left(x,y\right)\\ \sigma^{2}=\frac{1}{N_{x}N_{y}}\sum_{x,y}{\left(S_{m}\left(x,y\right)-\mu \right)}^{2}\\ \xi =\frac{1}{N_{x}N_{y}\sigma^{3}}\sum_{x,y}{\left(S_{m}\left(x,y\right)-\mu \right)}^{3}\\ \kappa =\frac{1}{N_{x}N_{y}\sigma^{4}}\sum_{x,y}{\left(S_{m}\left(x,y\right)-\mu \right)}^{4}-3 \end{array}$$

From Equation (Equation 24), we can build the first feature vector $Z_{s}$ of $N_{m}\times N_{bd}\times n=$ 4 × 3 × 3 = 36 elements, where $N_{m},N_{bd}$, and *n* are the number of moments, sub-bands, and scales. The feature vector $Z_{s}$ is represented as follows:$$Z_{s}=\left[Z_{1},Z_{2},Z_{3}\right]$$
where:$$Z_{1}=[\mu_{H_{1}},\mu_{V_{1}},\mu_{D_{1}}|\sigma_{H_{1}},\sigma_{V_{1}},\sigma_{D_{1}}|\xi_{H_{1}},\xi_{V_{1}},\xi_{D_{1}}\left|\kappa_{H_{1}},\kappa_{V_{1}},\kappa_{D_{1}}\right]$$
$$Z_{2}=[\mu_{H_{2}},\mu_{V_{2}},\mu_{D_{2}}|\sigma_{H_{2}},\sigma_{V_{2}},\sigma_{D_{2}}|\xi_{H_{2}},\xi_{V_{2}},\xi_{D_{2}}\left|\kappa_{H_{2}},\kappa_{V_{2}},\kappa_{D_{2}}\right]$$
$$Z_{3}=[\mu_{H_{3}},\mu_{V_{3}},\mu_{D_{3}}|\sigma_{H_{3}},\sigma_{V_{3}},\sigma_{D_{3}}|\xi_{H_{3}},\xi_{V_{3}},\xi_{D_{3}}\left|\kappa_{H_{3}},\kappa_{V_{3}},\kappa_{D_{3}}\right]$$

The second set of statistics is based on the prediction errors of coefficients $S_{m}\left(x,y\right)$ of an optimal linear predictor. The sub-band coefficients are correlated with their spatial, orientation, and scale neighbors. Several prediction techniques of coefficients $S_{H_{m}}^{p}\left(x,y\right)$, $S_{V_{m}}^{p}\left(x,y\right)$, and $S_{D_{m}}^{p}\left(x,y\right)$ (*m* = 1, 2, 3) may be used. In this work, we used a linear predictor, specifically the one proposed by Farid in [30], as shown below:(25)$$\begin{array}{c} S_{H_{m}}^{p}\left(x,y\right)=w_{1}S_{H_{m}}\left(x-1,y\right)+w_{2}S_{H_{m}}\left(x+1,y\right)+w_{3}S_{H_{m}}\left(x,y-1\right)+\\ \;w_{4}S_{H_{m}}\left(x,y+1\right)+w_{5}S_{H_{m+1}}\left(\frac{x}{2},\frac{y}{2}\right)+w_{6}S_{D_{m}}\left(x,y\right)+\\ \;w_{7}S_{D_{m+1}}\left(\frac{x}{2},\frac{y}{2}\right) \end{array}$$
(26)$$\begin{array}{c} S_{V_{m}}^{p}\left(x,y\right)=w_{1}S_{V_{m}}\left(x-1,y\right)+w_{2}S_{V_{m}}\left(x+1,y\right)+w_{3}S_{V_{m}}\left(x,y-1\right)+\\ \;w_{4}S_{V_{m}}\left(x,y+1\right)+w_{5}S_{V_{m+1}}\left(\frac{x}{2},\frac{y}{2}\right)+w_{6}S_{D_{m}}\left(x,y\right)+\\ \;w_{7}S_{D_{m+1}}\left(\frac{x}{2},\frac{y}{2}\right) \end{array}$$
(27)$$\begin{array}{c} S_{D_{m}}^{p}\left(x,y\right)=w_{1}S_{D_{m}}\left(x-1,y\right)+w_{2}S_{D_{m}}\left(x+1,y\right)+w_{3}S_{D_{m}}\left(x,y-1\right)+\\ \;w_{4}S_{D_{m}}\left(x,y+1\right)+w_{5}S_{D_{m+1}}\left(\frac{x}{2},\frac{y}{2}\right)+w_{6}S_{H_{m}}\left(x,y\right)+\\ \;w_{7}S_{V_{m+1}}\left(\frac{x}{2},\frac{y}{2}\right) \end{array}$$

For more clarity, in Figure 14, we provide the block diagram for the prediction of coefficient $S_{V_{1}}^{p}\left(x,y\right)$.

The parameters $w_{i}$ (scalar weighting values) of the error prediction coefficients of each sub-band for a given level *m* are adjusted to minimize the prediction error by minimizing the quadratic error function, as shown below:(28)$$E\left(w\right)={\left[S_{m}-Qw\right]}^{2}$$

The columns of the matrix *Q* contain the neighboring coefficient magnitudes, as specified in Equations (Equation 25)–(Equation 27). The quadratic error function is minimized analytically as follows:(29)$$\frac{dE(w)}{dw}=2Q^{T}\left(S_{m}-Qw\right)=0$$

Then, we obtain:(30)$$w_{opt}={\left(Q^{t}Q\right)}^{-1}Q^{t}S_{m}$$

For the optimal predictor, we use the log error given by the following equation to predict error coefficients of each sub-band for a given level *m*:(31)$$\epsilon_{m}^{p}=\log_{2}S_{m}-\log_{2}(|Qw_{opt}\left|\right)$$

By using Equation (Equation 31), additional statistics are collected, namely the mean, variance, skewness, and kurtosis (see Equation (Equation 24)). The feature vector $Z_{\epsilon }^{p}$ is similar to $Z_{s}$; it is represented as follows:$$Z_{\epsilon }^{p}=\left[Z_{1\epsilon }^{p},Z_{2\epsilon }^{p},Z_{3\epsilon }^{p}\right]$$
where:$$Z_{1\epsilon }^{p}=[\mu_{\epsilon_{H_{1}}}^{p},\mu_{\epsilon_{V_{1}}}^{p},\mu_{\epsilon_{D_{1}}}^{p}|\sigma_{\epsilon_{H_{1}}}^{p},\sigma_{\epsilon_{V_{1}}}^{p},\sigma_{\epsilon_{D_{1}}}^{p}|\xi_{\epsilon_{H_{1}}}^{p},\xi_{\epsilon_{V_{1}}}^{p},\xi_{\epsilon_{D_{1}}}^{p}\left|\kappa_{\epsilon_{H_{1}}}^{p},\kappa_{\epsilon_{V_{1}}}^{p},\kappa_{\epsilon_{D_{1}}}^{p}\right]$$
$$Z_{2\epsilon }^{p}=[\mu_{\epsilon_{H_{2}}}^{p},\mu_{\epsilon_{V_{2}}}^{p},\mu_{\epsilon_{D_{2}}}^{p}|\sigma_{\epsilon_{H_{2}}}^{p},\sigma_{\epsilon_{V_{2}}}^{p},\sigma_{\epsilon_{D_{2}}}^{p}|\xi_{\epsilon_{H_{2}}}^{p},\xi_{\epsilon_{V_{2}}}^{p},\xi_{\epsilon_{D_{2}}}^{p}\left|\kappa_{\epsilon_{H_{2}}}^{p},\kappa_{\epsilon_{V_{2}}}^{p},\kappa_{\epsilon_{D_{2}}}^{p}\right]$$
$$Z_{3\epsilon }^{p}=[\mu_{\epsilon_{H_{3}}}^{p},\mu_{\epsilon_{V_{3}}}^{p},\mu_{\epsilon_{D_{3}}}^{p}|\sigma_{\epsilon_{H_{3}}}^{p},\sigma_{\epsilon_{V_{3}}}^{p},\sigma_{\epsilon_{D_{3}}}^{p}|\xi_{\epsilon_{H_{3}}}^{p},\xi_{\epsilon_{V_{3}}}^{p},\xi_{\epsilon_{D_{3}}}^{p}\left|\kappa_{\epsilon_{H_{3}}}^{p},\kappa_{\epsilon_{V_{3}}}^{p},\kappa_{\epsilon_{D_{3}}}^{p}\right]$$

Finally, the feature vector that will be used for the learning classifier is represented by $Z=[Z_{s}|Z_{\epsilon }^{p}]$. It contains 72 components.

#### 5.2.2. Method 2: Feature Vectors Extracted from Empirical Moments of CF-Based Multi-Resolution

The first set of feature vectors $Z_{s}$ is extracted based on the CF and the wavelet decomposition, as proposed by Shi et al. [31]. The statistical moments of the characteristic function ϕ(k) of order n= 1 to 3 are represented for each sub-band $\left(A_{m},H_{m},V_{m},D_{m}\right)$ at different levels *m* = 1, 2, and 3 of the wavelet decomposition as follows:(32)$$M_{S_{m}}^{n}=\frac{\sum_{k=1}^{\frac{N}{2}}\left|\phi \left(k\right)\right|\times k^{n}}{\sum_{k=1}^{\frac{N}{2}}\left|\phi \left(k\right)\right|}$$
(33)$$\phi \left(k\right)=\sum_{i=1}^{N}h\left(i\right)\exp\left\{\frac{j2\pi ik}{K}\right\}\;1\leq k\leq K$$
is a component of the characteristic function at frequency *k*, calculated from the histogram of the sub-band $S_{m}$, and *N* is the total number of points of the histogram. Equation (Equation 32) allows us to build the first feature vector $Z_{m}$ of size 12 × 3 = 36 components and 3 moments of the initial image. The feature vectors $Z_{m}$ have been listed as follows:$$\begin{array}{cc} Z_{s}= & [M_{I}^{1},M_{I}^{2},M_{I}^{3}|M_{A_{1}}^{1},M_{A_{1}}^{2},M_{A_{1}}^{3}|M_{H_{1}}^{1},M_{H_{1}}^{2},M_{H_{1}}^{3}|M_{V_{1}}^{1},M_{V_{1}}^{2},M_{V_{1}}^{3}\\ & |M_{D_{1}}^{1},M_{D_{1}}^{2},M_{D_{1}}^{3}|M_{A_{2}}^{1},M_{A_{2}}^{2},M_{A_{2}}^{3}\left|M_{H_{2}}^{1},M_{H_{2}}^{2},M_{H_{2}}^{3}\right|M_{V_{2}}^{1},M_{V_{2}}^{2},M_{V_{2}}^{3}\\ & |M_{D_{2}}^{1},M_{D_{2}}^{2},M_{D_{2}}^{3}|M_{A_{3}}^{1},M_{A_{3}}^{2},M_{A_{3}}^{3}\left|M_{H_{3}}^{1},M_{H_{3}}^{2},M_{H_{3}}^{3}\right|M_{V_{3}}^{1},M_{V_{3}}^{2},M_{V_{3}}^{3}\\ & |M_{D_{3}}^{1},M_{D_{3}}^{2},M_{D_{3}}^{3}] \end{array}$$

In the above equation, $M_{I}^{1},M_{I}^{2},M_{I}^{3}$ are the moments of the initial image.

The second category of features is calculated from the moments of prediction-error image and its wavelet decomposition.

Prediction-error image:

In steganalysis, we only care about the distortion caused by data-hiding. This type of distortion may be rather weak and, hence, covered by other types of noises, including those caused due to the peculiar feature of the image itself. To make the steganalysis more effective, it is necessary to keep the noise of the dissimulation and eliminate most of the other noises. For this purpose, we calculate the moments of characteristic functions of order n= 1, 3 of the predicted error image and of its wavelet decomposition at the various levels m= 1, 2, and 3 (see Equation (Equation 32)). The prediction-error image is obtained by subtracting the predicted image (in which each predicted pixel grayscale value in the cover image uses its neighboring pixels’ grayscale values (see Equation (Equation 34))) from the cover image. Such features make the steganalysis more efficient because the hidden data is usually unrelated to the cover media. The prediction pixel is expressed as follows:(34)$$\hat{x}=\left\{\begin{array}{c} max(a,b)\;c\leq min(a,b)\\ min(a,b)\;c\geq max(a,b)\\ a+b-c\;otherwise \end{array}\right.$$

In the above equation, *a*, *b*, *c* are the context of the pixel *x* under consideration; $\hat{x}$ is the prediction value of *x*. The location of *a*, *b*, *c* can be illustrated as in Figure 15.

The feature vector $Z_{\epsilon }^{p}$ is represented as follows:$$\begin{array}{cc} Z_{\epsilon }^{p}= & [M_{\epsilon_{1}}^{p_{1}},M_{\epsilon_{1}}^{p_{2}},M_{\epsilon_{1}}^{p_{3}}|M_{A_{1}}^{1},M_{A_{1}}^{2},M_{A_{1}}^{3}|M_{H_{1}}^{1},M_{H_{1}}^{2},M_{H_{1}}^{3}|M_{V_{1}}^{1},M_{V_{1}}^{2},M_{V_{1}}^{3}\\ & |M_{D_{1}}^{1},M_{D_{1}}^{2},M_{D_{1}}^{3}|M_{A_{2}}^{1},M_{A_{2}}^{2},M_{A_{2}}^{3}\left|M_{H_{2}}^{1},M_{H_{2}}^{2},M_{H_{2}}^{3}\right|M_{V_{2}}^{1},M_{V_{2}}^{2},M_{V_{2}}^{3}\\ & |M_{D_{2}}^{1},M_{D_{2}}^{2},M_{D_{2}}^{3}|M_{A_{3}}^{1},M_{A_{3}}^{2},M_{A_{3}}^{3}\left|M_{H_{3}}^{1},M_{H_{3}}^{2},M_{H_{3}}^{3}\right|M_{V_{3}}^{1},M_{V_{3}}^{2},M_{V_{3}}^{3}\\ & |M_{D_{3}}^{1},M_{D_{3}}^{2},M_{D_{3}}^{3}] \end{array}$$

In the above equation, $M_{A_{1}}^{1},M_{A_{1}}^{2},M_{A_{1}}^{3}$ are the 1st, 2nd , and 3rd order moments of the corresponding CFs, from the sub-band $A_{1}$ of the 1^*st*^ level decomposition on the error image.

Finally, the feature vector that will be used for learning classification is $Z=[Z_{s}|Z_{\epsilon }^{p}]$, containing 78 components.

#### 5.2.3. Method 3: Feature Vector Extracted from Empirical Moments Based on the FC and the PDF of Image Prediction Error and Its Different Sub-Bands of the Multi-Resolution Decomposition

The first characteristic vector $Z_{s}$ combines two types of normalized moments: moments based on the function density of probability and moments based on the characteristic function of various sub-bands of the multi-resolution decomposition at three levels of the gray image. We use the expression of Wang and Moulin [32] to calculate the moments of order n= 1 to 6 of the initial image and its sub-band $\left(A_{m},H_{m},V_{m},D_{m}\right)$ of the three-level (m= 1 to 3) wavelet decomposition, as shown below:(35)$$M_{S_{m}}^{n}=\frac{\sum_{k=1}^{\frac{N}{2}}\left|\phi \left(k\right)\right|\times \sin^{n}\left(\frac{\pi k}{K}\right)}{\sum_{k=1}^{\frac{N}{2}}\left|\phi \left(k\right)\right|}$$
(36)$$\phi \left(k\right)=\sum_{i=1}^{N}h\left(i\right)\exp\left\{\frac{j2\pi ik}{K}\right\}\;1\leq k\leq K$$
is a component of the characteristic function at frequency k, estimated from the histogram. Equation (Equation 35) already allows having a feature vector of 6 × 1 + 6 × (4 × 3) = 78 components. Also, to improve the performance of the learning system, we calculate the moments of the sub-bands $A_{2}^{{}^{\prime }}$, $H_{2}^{{}^{\prime }}$, $V_{2}^{{}^{\prime }}$, $D_{2}^{{}^{\prime }}$ obtained from the decomposition of the diagonal sub-band $D_{1}$. Therefore, the total size of the vector $Z_{s}$ is 78 + (6 × 4) = 102 components.
$$\begin{array}{cc} Z_{s}= & [M_{I}^{i}|M_{A_{1}}^{i}|M_{H_{1}}^{i}|M_{V_{1}}^{i}|M_{D_{1}}^{i}|M_{A_{2}}^{i}|M_{H_{2}}^{i}|M_{V_{2}}^{i}|M_{D_{2}}^{i}|M_{A_{3}}^{i}|M_{H_{3}}^{i}|M_{V_{3}}^{i}|M_{D_{3}}^{i}\\ & |M_{A_{2}^{{}^{\prime }}}^{i}|M_{H_{2}^{{}^{\prime }}}^{i}|M_{V_{2}^{{}^{\prime }}}^{i}\left|M_{D_{2}^{{}^{\prime }}}^{i}\right],\;i=1,2,\ldots ,6 \end{array}$$

For example, $M_{I}^{i}=\left[M_{I}^{1},M_{I}^{2},M_{I}^{3},M_{I}^{4},M_{I}^{5},M_{I}^{6}\right]$ are the first six order moments of the original image.

The second category of characteristics consists of the first six moments of the prediction error, which is $\epsilon_{m}^{p}=\log_{2}S_{m}-\log_{2}(|Qw_{opt}\left|\right)$ of coefficients of each sub-band for a given level *m*, as shown below:(37)$$m_{\epsilon_{m}^{p}}^{n}=\frac{1}{N}\sum_{i=1}^{N}{\left(\epsilon_{m}^{p}\right)}^{n}\;n=1,2.\ldots ,6$$

The vector of the second category is defined by $Z_{\epsilon }^{p}$, as shown below:$$Z_{\epsilon }^{p}=[m_{\epsilon H_{m}}^{i}|m_{\epsilon V_{m}}^{i}\left|m_{\epsilon D_{m}}^{i}\right]$$
for each
$$m=\left\{1,2,3\right\};i=1,2,\ldots ,6$$

The size of $Z_{\epsilon }^{p}$ is 3 x 6 x 3 = 54 components.

Finally, the feature vector to be used for classification by learning is $Z=[Z_{s}|Z_{\epsilon }^{p}]$. It has 156 components.

### 5.3. Classification

The last stage of the learning and test process of the universal steganalysis is classification (see Figure 12). Its objective is to group the images into two classes, class of the cover images and class of the stego images, according to their feature vectors. We adopt the Fisher linear discriminator (FLD) and the support vector machine (SVM) for training and testing.

#### 5.3.1. FLD Classifier

Below, we reformulate the FLD classifier for our application and apply it to two classes. Let $Z=\left\{Z_{1},Z_{2},\ldots ,Z_{N}\right\}$ be a set of feature vectors, each with nd dimensions. Among these vectors, $N_{1}$ vectors are $Z_{c}$ feature vectors labeled 1, indicating cover images. $N_{2}$ vectors are $Z_{s}$ labeled 2, indicating stego images, with $N=N_{1}+N_{2}$. We want to form all projection values $(Z_{p})=\left\{Z_{p_{1}},Z_{p_{2}},\ldots ,Z_{p_{N}}\right\}$ of dimension *N* through linear combinations of feature vectors $Z_{p}$ as follows:(38)$$Z_{p}=W^{t}Z$$

In the above equation, W is an orientation vector of dimension nd.

In our study, the feature vector Z is projected into a space of two classes. This projection tends to maximize the distance between the projected class means $\left(M_{cp},M_{sp}\right)$ while minimizing projected class scatters $S_{cp},S_{sp}$.

Learning processThe learning process involves optimizing the following expression:
(39)$$J\left(W\right)=\frac{|M_{cp}-M_{sp}{|}^{2}}{S_{cp}+S_{sp}}$$
where:
(40)$$M_{cp}=\frac{1}{N_{1}}\sum_{Z_{p}\in Z_{cp}}Z_{p}=\frac{1}{N_{1}}\sum_{Z\in Z_{c}}W^{t}Z=W^{t}M_{c}$$
is the mean feature vector of cover class after projection, and
(41)$$M_{c}=\frac{1}{N_{1}}\sum_{Z\in Z_{c}}Z$$
is the mean feature vector of cover class of dimension nd.The mean feature vector of stego class after projection is represented as follows:
(42)$$M_{sp}=\frac{1}{N_{2}}\sum_{Z_{p}\in Z_{sp}}Z_{p}=\frac{1}{N_{2}}\sum_{Z\in Z_{s}}W^{t}Z=W^{t}M_{s}$$
where:
(43)$$M_{s}=\frac{1}{N_{2}}\sum_{Z\in Z_{s}}Z$$
is the mean feature vector of a stego class of dimension nd.The scatter matrix of the cover class after projection has been shown as follows:
(44)$$S_{cp}=\sum_{Z_{p}\in Z_{cp}}{\left(Z_{p}-M_{cp}\right)}^{2}=\sum_{Z\in Z_{c}}{\left(W^{t}Z-W^{t}M_{c}\right)}^{2}=\sum_{Z\in Z_{c}}W^{t}\left(Z-M_{c}\right){\left(Z-M_{c}\right)}^{t}W=W^{t}S_{c}W$$
where:
(45)$$S_{c}=\left(Z-M_{c}\right){\left(Z-M_{c}\right)}^{t}$$
is the scatter matrix (of dimension nd×nd) of a cover class.The scatter matrix of the projected samples of a stego class has been shown as follows:
(46)$$S_{sp}=\sum_{Z_{p}\in Z_{sp}}{\left(Z_{p}-M_{sp}\right)}^{2}=\sum_{Z\in Z_{s}}{\left(W^{t}Z-W^{t}M_{s}\right)}^{2}=\sum_{Z\in Z_{s}}W^{t}\left(Z-M_{s}\right){\left(Z-M_{s}\right)}^{t}W=W^{t}S_{s}W$$
where:
(47)$$S_{s}=\left(Z-M_{s}\right){\left(Z-M_{s}\right)}^{t}$$
is a scatter matrix (of dimension nd×nd) for the samples in the original feature space of a stego class.The within-class scatter matrix after projection is defined as follows:
(48)$$S_{cp}+S_{sp}=W^{t}\left(S_{c}+S_{s}\right)W=W^{t}S_{w}W$$
where:
(49)$$S_{w}=S_{c}+S_{s}$$
The difference between the projected means is expressed as follows:
(50)$${\left(M_{cp}-M_{sp}\right)}^{2}={\left(W^{t}M_{c}-W^{t}M_{s}\right)}^{2}=W^{t}\left(M_{c}-M_{s}\right){\left(M_{c}-M_{s}\right)}^{t}W=W^{t}S_{B}W$$
where:
(51)$$S_{B}=\left(M_{c}-M_{s}\right){\left(M_{c}-M_{s}\right)}^{t}$$
We can finally express the Fisher criterion (Equation (Equation 39)) in terms of $S_{B}$ and $S_{W}$ as follows:
(52)$$J\left(W\right)=\frac{W^{t}S_{B}W}{W^{t}S_{w}W}$$
The solution of Equation (Equation 52) is given by [49].
(53)$$W_{opt}=S_{w}^{-1}\left(M_{c}-M_{s}\right)$$Testing processThe testing process (classification step) is conducted as follows:Let *Z* be the matrix containing the feature vectors of covers and stegos.The projection of *Z* on the orientation vector $W_{opt}$ gives all projected values $Z_{p}$.
(54)$$Z_{p}\left(j\right)=\sum_{i=1}^{9}W_{opt}\left(i\right)\times Z\left(i,j\right)+b\;j=1,2,\ldots ,N$$
*b* is a threshold of discrimination between both classes, and it can be fixed to a value that is halfway between both averages projected on the cover and stego.
(55)$$b=0.5\times (M_{cp}+M_{sp})$$
with:
$$\begin{array}{c} M_{cp}=W_{opt}^{t}\times M_{c}\\ M_{sp}=W_{opt}^{t}\times M_{s} \end{array}$$In the above equations, $W_{opt}^{t}$ is the transposed of $W_{opt}$.The result $Z_{p}\left(j\right)$, j=1,…,N determines the cover or stego class of every test image.Indeed, if $Z_{p}\left(j\right)\geq 0$, then the image under test is cover; otherwise, it is stego.

#### 5.3.2. SVM Classifier

According to numerous recent studies, the SVM classification method is better than the other data classification algorithms in terms of classification accuracy [50]. SVM performs classification by creating a hyper-plan that separates the data into two categories in the most optimal way.

Let ${\left(Z_{i},y_{i}\right)}_{\left(1\leq i\leq N\right)}$ be a set of training examples, each example $Z_{i}\in R^{nd}$, nd being the dimension of the input space; it belongs to a class labeled as $y_{i}\in \left\{-1,1\right\}$. SVM classification constructs a hyper-plan $W^{T}Z+b=0$, which best separates the data through a minimizing process, as shown below:(56)$$\begin{array}{c} \frac{1}{2}{∥w∥}^{2}+C\sum_{i=1}^{N}\zeta_{i}\\ \;subject\;to:y_{i}\left(wZ_{i}+b\right)\geq 1-\zeta_{i} \end{array}$$

Variables $\zeta_{i}$ are called slack variables, and they measure the error made at point $\left(Z_{i},y_{i}\right)$.

Parameter *C* can be viewed as a way to control overfitting.

$\zeta_{i}\geq 0$ and C>0 is the trade-off between regularization and constraint violation.

Problems related to quadratic optimization are a well-known class of mathematical programming problems, and many (rather intricate) algorithms exist to aid in solving them. Solutions involve constructing a dual problem where a Lagrange multiplier $\alpha_{i}$ is associated with every constraint in the primary problem, as shown below:(57)$$\begin{array}{c} L\left(\alpha \right)=\sum_{i}\alpha_{i}-\frac{1}{2}\sum_{i}\sum_{j}\alpha_{i}\alpha_{j}y_{i}y_{j}Z_{i}^{T}Z_{j}\\ \;subject\;to:\sum_{i}\alpha_{i}y_{i}=0\\ \;0\leq \alpha_{i}y_{i}\leq C \end{array}$$

$\alpha_{i}$ or Lagrange multipliers are also known as support values.

The linear classifier presented previously is very limited. In most case, classes not only overlap, but the genuine separation functions are non-linear hyper-surfaces. The motivation for such an extension is that an SVM that can create a non-linear decision hyper-surface will be able to non-linearly classify separable data.

The idea is that the input space can always be mapped on to a higher dimensional feature space where the training set is separable.

The linear classifier relies on the dot product between vectors $K\left(Z_{i},Z_{j}\right)=Z_{i}^{T}Z_{j}$. If every data point is mapped on to a high-dimensional space via some transformation Φ:Z→φ(Z), the dot product becomes:$K\left(Z_{i},Z_{j}\right)=\varphi {\left(Z_{i}\right)}^{T}\varphi \left(Z_{j}\right)$. Then in the dual formulation, we maximize the following:(58)$$\begin{array}{c} L\left(\alpha \right)=\sum_{i=1}^{N}\alpha_{i}-\frac{1}{2}\sum_{i}\sum_{j}\alpha_{i}\alpha_{j}y_{i}y_{j}K\left(Z_{i},Z_{j}\right)\\ \;subject\;to:\sum_{i}\alpha_{i}y_{i}=0\\ \;0\leq \alpha_{i}y_{i}\leq C \end{array}$$

Subsequently, the decision function turns into the following:(59)$$f\left(x\right)=sgn(\sum_{i=n}^{m}\alpha_{i}y_{i}K\left(Z_{i},Z\right)+b)$$

It should be noted that the dual formulation only requires access to the kernel function and not the features Φ(.), allowing one to solve the formulation in very high-dimensional feature spaces efficiently. This is also called the kernel trick.

There are many kernel functions in SVM. Therefore, determining how to select a good kernel function is also a research issue. However, for general purposes, there are some popular kernel functions [50,51], which have been listed as follows:Linear Kernel:
(60)$$K\left(Z_{i},Z_{j}\right)=Z_{i}^{T}Z_{j}$$Polynomial Kernel:
(61)$$K\left(Z_{i},Z_{j}\right)={\left(\gamma Z_{i}^{T}Z_{j}+r\right)}^{d}\;\gamma >0$$RBF Kernel:
(62)$$K\left(Z_{i},Z_{j}\right)=\exp\left(-\gamma {∥Z_{i}-Z_{j}∥}^{2}\right)\;\gamma >0$$Sigmoid Kernel:
(63)$$K\left(Z_{i},Z_{j}\right)=\tan h\left(\gamma Z_{i}^{T}Z_{j}+r\right)$$

Here, γ, *r*, and *d* are kernel parameters.

In our work, we used the RBF kernel function.

## 6. Experimental Results of Steganalysis

In this section, we present some experimental results that were obtained from the studied steganalysis system that was applied to the enhanced steganographic methods in the spatial and frequency domain. For this purpose, the image dataset UCID [52,53] is used, which includes 1338 uncompressed color images, and all the images were converted to grayscale before conducting the experiments.

In our experiments, we first created the stego images using the following steganographic methods: Enhanced EALSBMR (EEALSBMR), Enhanced DCT steganography (EDCT), and Enhanced DWT steganography (EDWT). We used these methods with different embedding rates of 5%, 10%, and 20%. Following this, we extracted the image features using the three feature-extraction techniques described above (Farid, Shi, and Moulin techniques) for both the cover and stego images. Finally, we employed the classifiers FLD and SVM to classify the images as either containing a hidden message or not. The evaluation of the classification (binary classification) and the steganalysis (also indirectly the efficiency of insertion methods) is performed by calculating the following parameters: sensibility, specificity, and precision of the confusion matrix and the Kappa coefficient (see Table 8 and Equation (Equation 64))
(64)$$Kappa=\frac{P_{0}-P_{a}}{1-P_{a}}$$
with:(65)$$P_{0}=TP+TN;P_{a}=\left(TP+FP\right)\times \left(TP+FN\right)+\left(FN+TN\right)\times \left(FP+TN\right)$$

In the above equation, $P_{0}$ is the total agreement probability (related to the accuracy), and $P_{a}$ is the agreement probability that arises out of chance.

Here is one possible interpretation of Kappa values:Poor agreement = Less than 0.20Fair agreement = 0.20 to 0.40Moderate agreement = 0.40 to 0.60Good agreement = 0.60 to 0.80Very good agreement = 0.80 to 1.00

### 6.1. Classification Results Applied to the Steganographic Method EEALSBMR

In Table 9, Table 10, Table 11, Table 12, Table 13 and Table 14, we present the classification results (steganalysis) based on the classifiers FLD and SVM and the features of Farid, Shi, and Moulin for the EEALSBMR insertion method with different insertion rates of 5%, 10%, and 20%. The results show that steganalysis is not effective for all insertion rates. Indeed, the values Se, Sp, and Pr vary around 50%, so these values are not informative values and do not give any idea about the nature of the data. The value of the Kappa coefficient (lower than 0.2) confirms these results. The EEALSBMR steganographic method is robust against statistical steganalysis techniques.

### 6.2. Classification Results Applied to the Steganographic Method EDCT

The classification results (steganalysis) provided in Table 15, Table 16, Table 17, Table 18, Table 19 and Table 20 for the EDCT insertion method show that with the FLD classifier, when the insertion rate is equal to or higher than 20%, steganalysis is very effective with Shi features and Moulin features, but it is less effective with Farid features. With the SVM classifier, except in the case of Shi features, when an insertion rate of 20% is applied, the results obtained are quite similar to those obtained from the EEALSBMR algorithm and, therefore, steganalysis is not effective. It should be noted that the FLD classifier is more effective for a feature vector of a high dimension than the SVM classifier.

### 6.3. Classification Results Applied to the Steganographic Method EDWT

With respect to the EDWT method, the results are provided in Table 21, Table 22, Table 23, Table 24, Table 25 and Table 26. These results obtained with the classifiers FLD and SVM indicate that the values of the parameters Se, Sp, Pr, Ac, and Kappa are high for all insertion rates and feature vectors (Farid, Shi, and Moulin). These results can easily inform us about the presence of hidden information; therefore, steganalysis can be concluded to be very effective. As a result, the insertion method is not robust. It should be noted that steganalysis is very effective here because both the steganagraphic method and feature vectors are based on multi-resolution wavelet decomposition.

### 6.4. Discussion

The enhanced adaptive LSB methods of steganography in the spatial domain (EEALSBMR) and frequency domain (EDCT and EDWT) provide stego images with a good visual quality up to an embedding rate of 40%: the PSNR is over 50 dB, and the distortion is not visible to the naked eye. Security of the message contents, in case detected by an opponent, is ensured by using the chaotic system. On the other hand, we applied a universal steganalysis method that can work well with all known and unknown steganography algorithms. Universal steganalysis methods exploit the changes in certain inherent features of the cover images when a message is embedded. The accuracy of the classification (discrimination between two classes: cover and stego) of the system greatly relies on several factors, such as the choice of the right characteristic vectors, the classifier, and its parameters.

## 7. Conclusions

In this work, we first improved the structure and security of three steganagraphic methods that are studied in the spatial and frequency domain by integrating them with a robust proposed chaotic system. Following this, we built a statistical steganalysis system to evaluate the robustness of the three enhanced steganographic methods. In this system, we selected three different feature vectors, namely higher-order statistics of high-frequency wavelet sub-bands and their prediction errors, statistical moments of the characteristic functions of the prediction-error image, the test image, and their wavelet sub-bands, and both empirical PDF moments and the normalized absolute CF. After this, we applied two types of classifiers, namely FLD and SVM, with the RBF kernel.

Extensive experimental work has demonstrated that the proposed steganalysis system based on the multi-dimensional feature vectors can detect hidden messages using the EDWT steganographic method, irrespective of the message size. However, it cannot distinguish between cover and stego images using the EEALSBMR steganographic and EDCT methods if the message size is smaller than 20% and 15%, respectively.

## Figures and Tables

**Figure 1 entropy-21-00748-f001:**
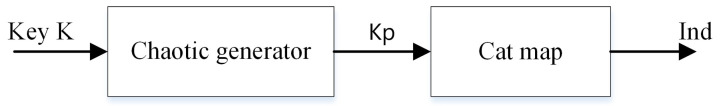
Proposed chaotic generator.

**Figure 2 entropy-21-00748-f002:**
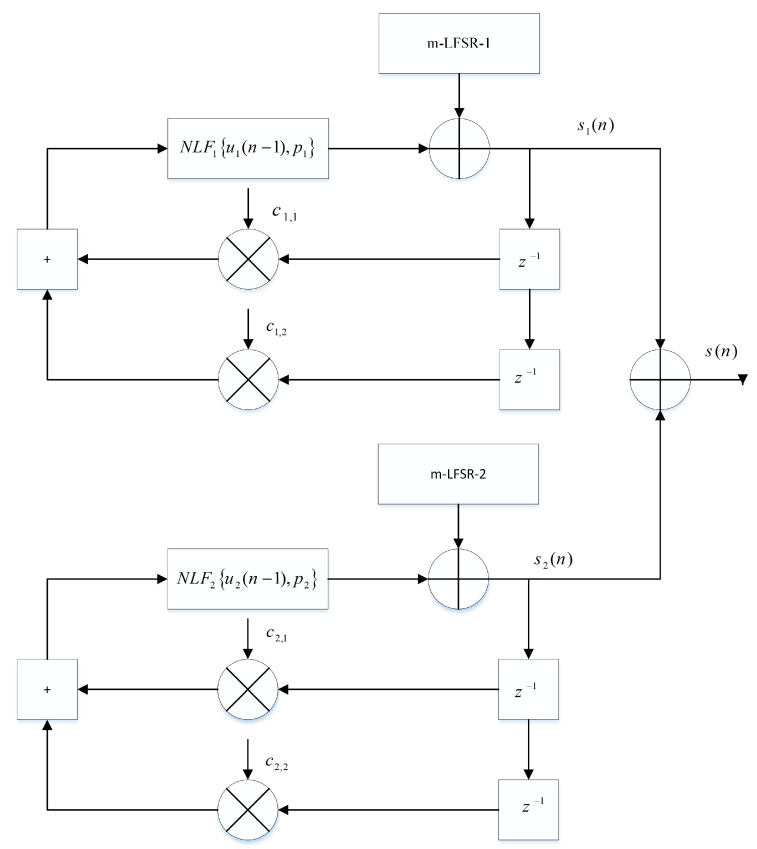
Chaotic generator.

**Figure 3 entropy-21-00748-f003:**
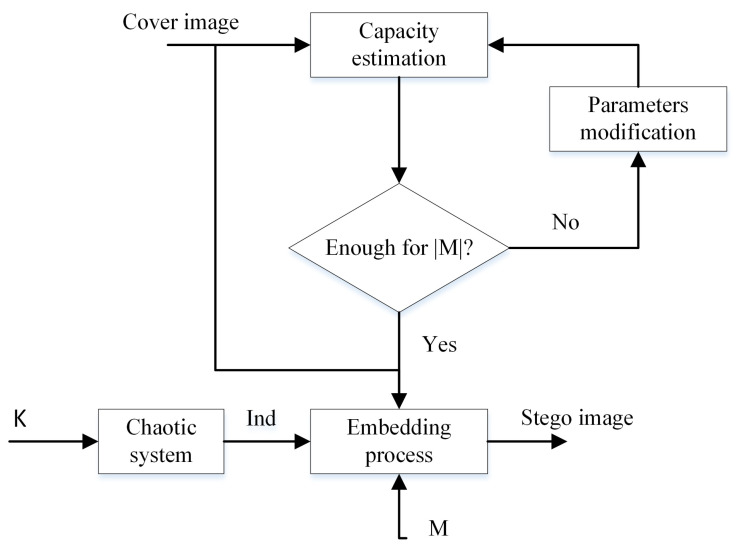
EEALSBMR insertion procedure.

**Figure 4 entropy-21-00748-f004:**
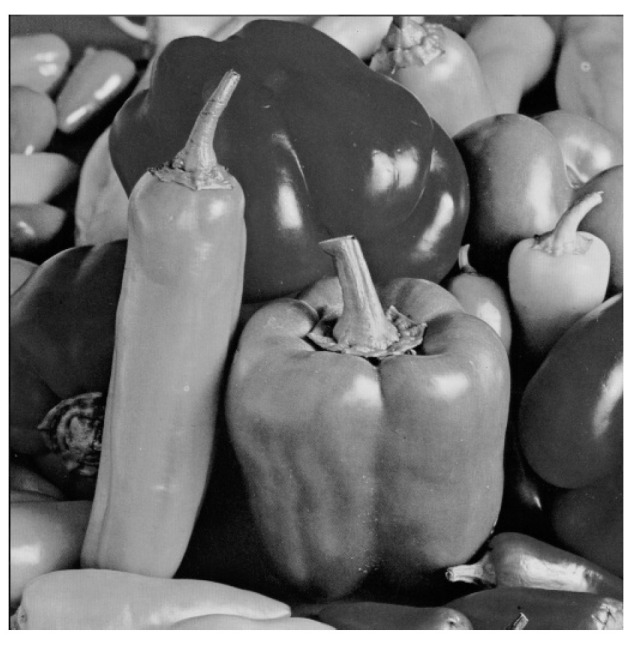
“Peppers” as cover image.

**Figure 5 entropy-21-00748-f005:**
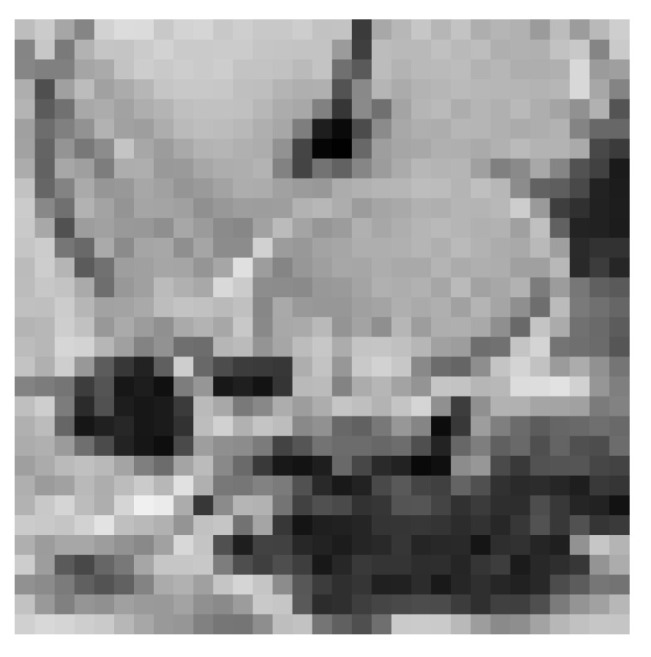
“Bike” is as embedded message.

**Figure 6 entropy-21-00748-f006:**
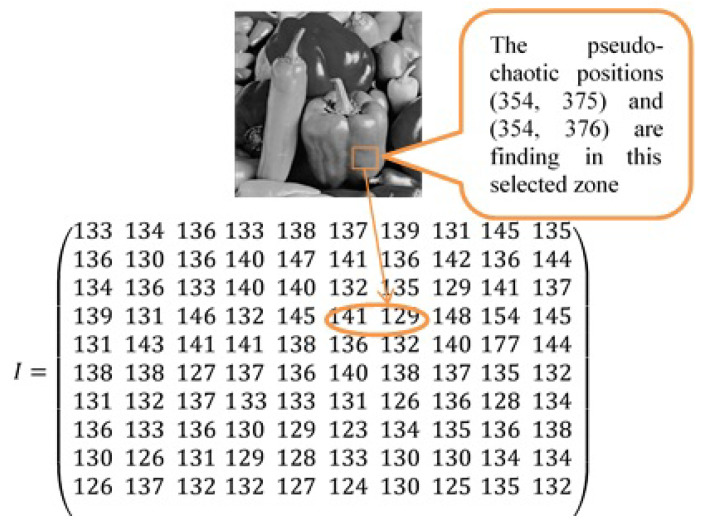
Pseudo-chaotic block selection and its corresponding gray value.

**Figure 7 entropy-21-00748-f007:**
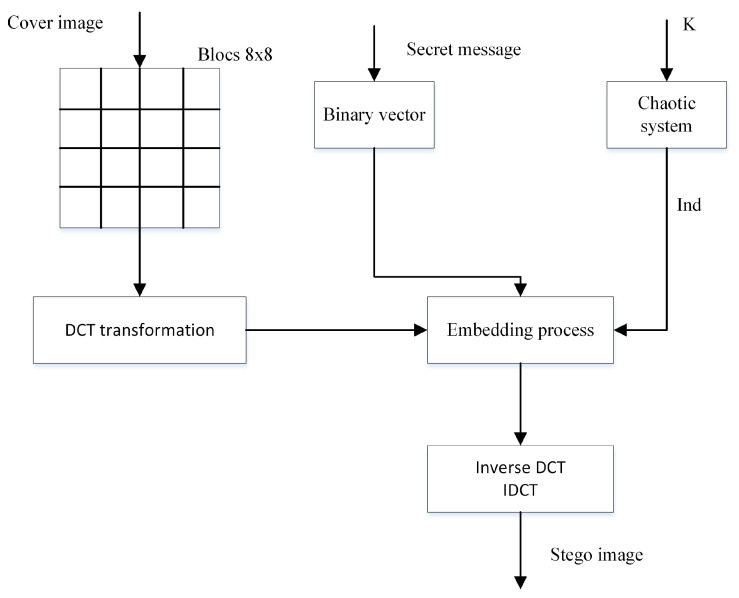
Diagram of the enhanced steganographic-based DCT transform.

**Figure 8 entropy-21-00748-f008:**
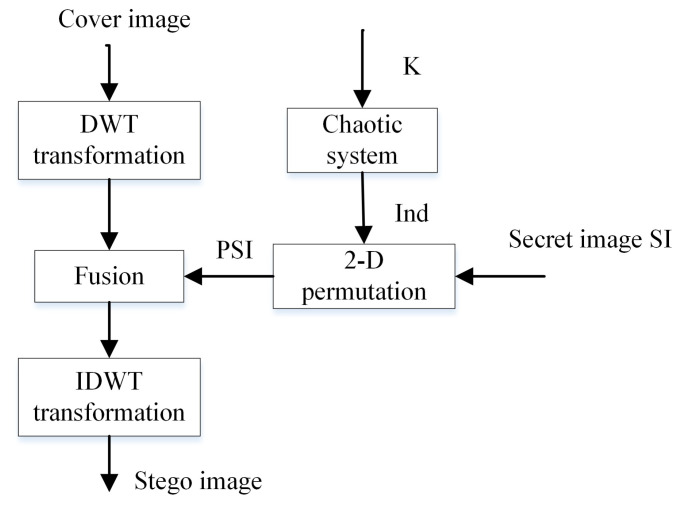
Diagram of the EDWT algorithm.

**Figure 9 entropy-21-00748-f009:**
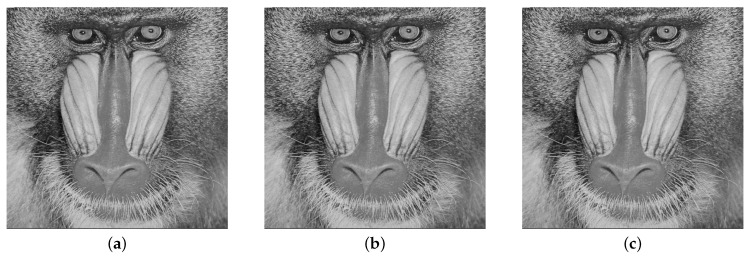
(**a**) Cover image, (**b**) Stego image with embedding rate of 5%, (**c**) Stego image with embedding rate of 40%.

**Figure 10 entropy-21-00748-f010:**
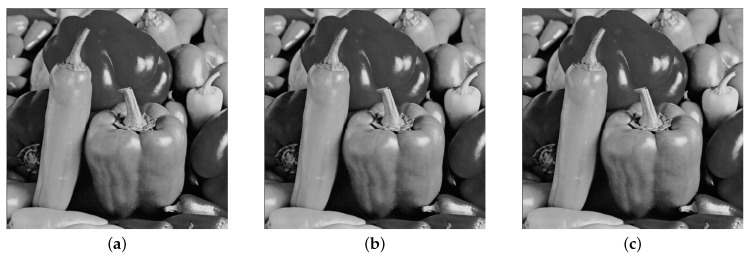
(**a**) Cover image, (**b**) Stego image with embedding rate of 5%, (**c**) Stego image with embedding rate of 40%.

**Figure 11 entropy-21-00748-f011:**
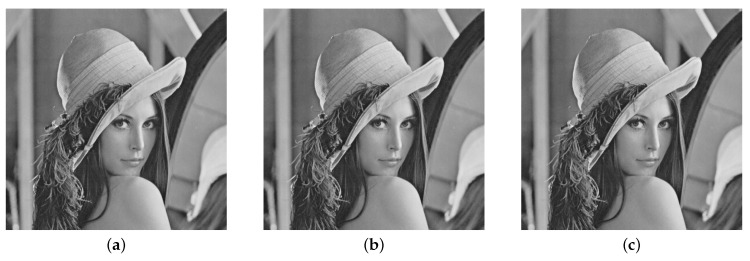
(**a**) Cover image, (**b**) Stego image with embedding rate of 5%, (**c**) Stego image with embedding rate of 40%.

**Figure 12 entropy-21-00748-f012:**
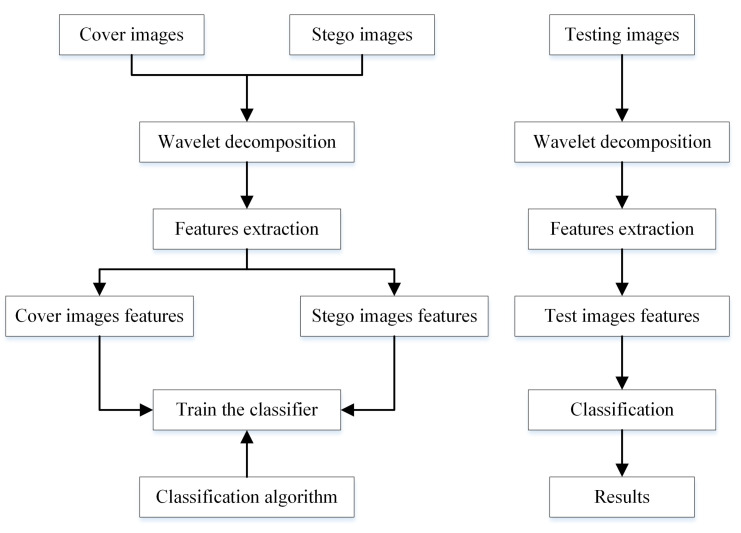
Flowchart of the blind steganalysis process.

**Figure 13 entropy-21-00748-f013:**
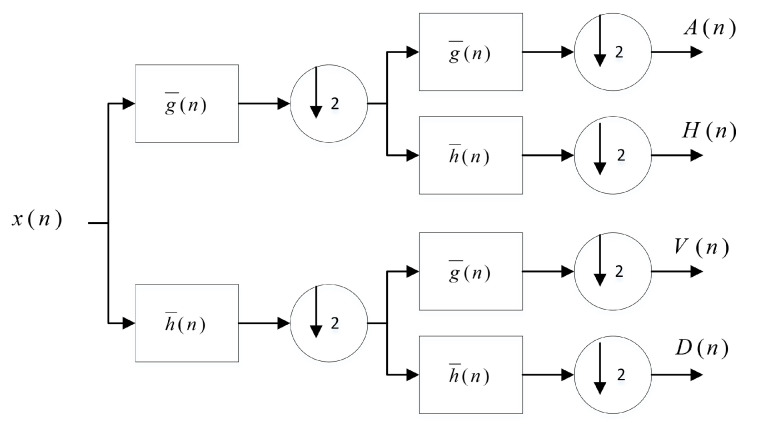
Multi-resolution wavelet decomposition.

**Figure 14 entropy-21-00748-f014:**
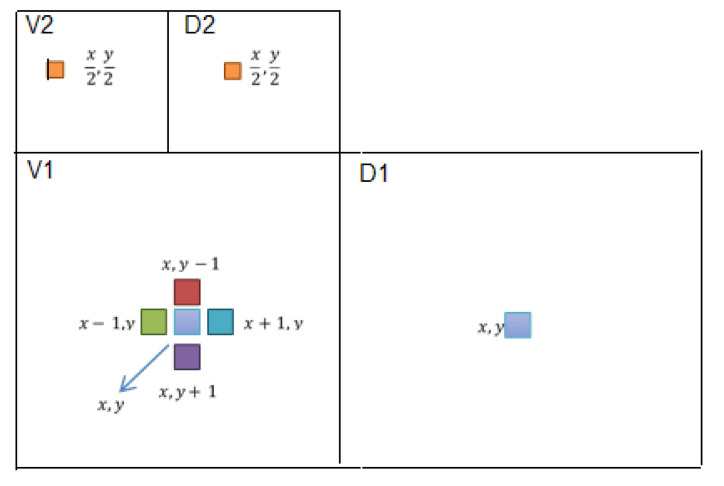
Block diagram for the prediction of coefficient $S_{V_{1}}^{p}\left(x,y\right)$.

**Figure 15 entropy-21-00748-f015:**
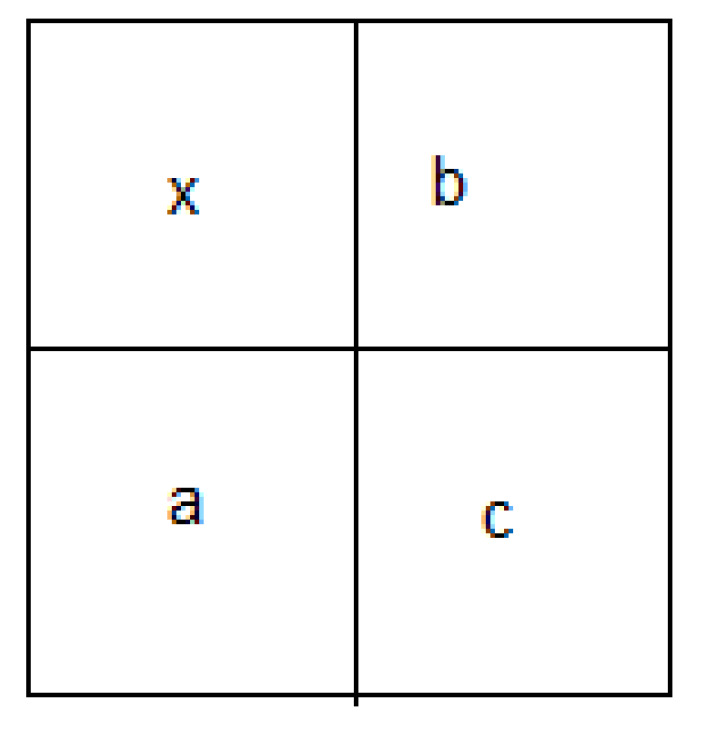
Prediction context of a pixel *x*.

**Table 1 entropy-21-00748-t001:** PSNR, IF, and SSIM values for the EEALSBMR method.

Embedding Rate	Cover Image	PSNR	IF	SSIM
5%	Baboon	68.3810	0.9999	0.9999
	Lena	68.1847	0.9999	0.9999
	Peppers	67.7160	0.9999	0.9999
10%	Baboon	65.5986	0.9999	0.9999
	Lena	65.2821	0.9999	0.9999
	Peppers	64.7763	0.9999	0.9999
20%	Baboon	62.3551	0.9999	0.9999
	Lena	62.3559	0.9999	0.9996
	Peppers	61.7066	0.9999	0.9995
30%	Baboon	60.6902	0.9998	0.9999
	Lena	60.5630	0.9998	0.9990
	Peppers	59.9585	0.9998	0.9992
40%	Baboon	59.4245	0.9997	0.9999
	Lena	59.2608	0.9997	0.9985
	Peppers	58.6662	0.9997	0.9988

**Table 2 entropy-21-00748-t002:** PSNR, IF, and SSIM values for the EDCT method.

Embedding Rate	Cover Image	PSNR	IF	SSIM
5%	Baboon	71.2372	0.9999	0.9999
	Lena	71.1769	0.9999	0.9999
	Peppers	70.4866	0.9999	0.9999
10%	Baboon	64.8846	0.9999	0.9999
	Lena	64.9487	0.9999	0.9998
	Peppers	64.1426	0.9999	0.9998
20%	Baboon	59.6895	0.9997	0.9999
	Lena	59.6225	0.9997	0.9992
	Peppers	58.9535	0.9997	0.9993
30%	Baboon	57.4212	0.9995	0.9998
	Lena	57.3421	0.9995	0.9989
	Peppers	56.7406	0.9995	0.9988
40%	Baboon	56.3421	0.9994	0.9997
	Lena	56.2265	0.79994	0.9987
	Peppers	55.4876	0.9994	0.9985

**Table 3 entropy-21-00748-t003:** PSNR, IF, and SSIM values for EDWT method.

Embedding Rate	Cover Image	PSNR	IF	SSIM
5%	Baboon	59.1876	0.9999	0.9999
	Lena	58.7673	0.9997	0.9999
	Peppers	58.1699	0.9997	0.9999
10%	Baboon	56.2224	0.9997	0.9999
	Lena	55.8085	0.9994	0.9999
	Peppers	55.2086	0.9993	0.9999
20%	Baboon	53.3463	0.9988	0.9999
	Lena	52.8205	0.9988	0.9999
	Peppers	52.2269	0.9987	0.9999
30%	Baboon	52.0465	0.9984	0.9999
	Lena	51.6471	0.9984	0.9999
	Peppers	51.0509	0.9983	0.9999
40%	Baboon	51.3450	0.9982	0.9999
	Lena	50.9536	0.9981	0.9999
	Peppers	50.3417	0.9980	0.9999

**Table 4 entropy-21-00748-t004:** *E*, *R*, and IR for the EEALSBMR method.

Embedding Rate	Cover Image	*E*	*R*	IR
5%	Baboon	7.3586	0.0802	0.3805
	Lena	7.4455	0.0693	0.3261
	Peppers	7.5715	0.0536	0.2975
10%	Baboon	7.3586	0.0802	0.3805
	Lena	7.4456	0.0693	0.3261
	Peppers	7.5715	0.0535	0.2976
20%	Baboon	7.3585	0.0802	0.3805
	Lena	7.4457	0.0693	0.3261
	Peppers	7.5717	0.0535	0.2977
30%	Baboon	7.3584	0.0802	0.3805
	Lena	7.4457	0.0693	0.3261
	Peppers	7.5718	0.0535	0.2975
40%	Baboon	7.3578	0.0803	0.3806
	Lena	7.4454	0,0693	0.3260
	Peppers	7.5722	0.0535	0.2973

**Table 5 entropy-21-00748-t005:** *E*, *R*, and IR values for the EDCT method.

Embedding Rate	Cover Image	*E*	*R*	IR
5%	Baboon	7.3585	0.0802	0.3804
	Lena	7.4456	0.0693	0.3261
	Peppers	7.5716	0.0536	0.2976
10%	Baboon	7.3585	0.0802	0.3805
	Lena	7.4456	0.0693	0.3262
	Peppers	7.5717	0.0535	0.2976
20%	Baboon	7.3585	0.0802	0.3804
	Lena	7.4457	0.0693	0.3263
	Peppers	7.5725	0.0534	0.2973
30%	Baboon	7.3584	0.0802	0.3802
	Lena	7.4459	0.0693	0.3261
	Peppers	7.5730	0.0534	0.2969
40%	Baboon	7.3578	0.0803	0.3806
	Lena	7.4462	0,0692	0.3257
	Peppers	7.5734	0.0533	0.2973

**Table 6 entropy-21-00748-t006:** *E*, *R*, and IR values for EDWT method.

Embedding Rate	Cover Image	*E*	*R*	IR
5%	Baboon	7.3581	0.0802	0.3805
	Lena	7.4455	0.0693	0.3261
	Peppers	7.5715	0.0536	0.2975
10%	Baboon	7.3580	0.0802	0.3806
	Lena	7.4456	0.0693	0.3261
	Peppers	7.5717	0.0535	0.2974
20%	Baboon	7.3580	0.0802	0.3806
	Lena	7.4456	0.0693	0.3261
	Peppers	7.5718	0.0535	0.2975
30%	Baboon	7.3580	0.0802	0.3805
	Lena	7.4456	0.0693	0.3261
	Peppers	7.5718	0.0535	0.2974
40%	Baboon	7.3580	0.0803	0.3806
	Lena	7.4457	0,0693	0.3261
	Peppers	7.5721	0.0533	0.2973

**Table 7 entropy-21-00748-t007:** *E*, *R*, and IR values for the cover images.

Cover Image	*E*	*R*	IR
Baboon	7.3585	0.0802	0.3805
Lena	7.4455	0.0693	0.3261
Peppers	7.5715	0.0536	0.2976

**Table 8 entropy-21-00748-t008:** Confusion matrix.

		H0: Stego Image	H1: Cover Image	
Test outcome	Test outcome positive	True Positive TP	False Positive FP	Positive predictive value (PPV), or Precision $Pr=\frac{TP}{TP+FP}$
	Test outcome negative	False Negative FN	True Negative TN	Negative predictive value (NPV) $NPV=\frac{TN}{TN+FN}$
		True positive rate (TPR), or, Sensitivity (Se), $Se=\frac{TP}{TP+FN}$	True negative rate (TNR), or Specificity(Sp), $Sp=\frac{TN}{TN+FP}$	Accuracy (Ac), $Ac=\frac{TP+TN}{TP+FN+FP+TN}$

**Table 9 entropy-21-00748-t009:** FLD classification evaluation of EEALSBMR algorithm using Farid features.

**5%**	**H0: Stego Images**	**H1: Cover Images**	
H0	0.2744	0.2714	Pr = 0.5027
H1	0.2256	0.2286	NPV = 0.5033
	Se = 0.5487	Sp = 0.4572	Ex = 0.5030
Kappa = 0.0060			
**10%**	**H0: Stego Images**	**H1: Cover Images**	
H0	0.2690	0.2645	Pr = 0.5042
H1	0.2310	0.2355	NPV = 0.5048
	Se = 0.5380	Sp = 0.4710	Ex = 0.5045
Kappa = 0.0090			
**20%**	**H0: Stego Images**	**H1: Cover Images**	
H0	0.2745	0.2459	Pr = 0.5275
H1	0.2255	0.2541	NPV = 0.5298
	Se = 0.5490	Sp = 0.5082	Ex = 0.5286
Kappa = 0.0572			

**Table 10 entropy-21-00748-t010:** FLD classification evaluation of EEALSBMR algorithm using Shi features.

**5%**	**H0: Stego Images**	**H1: Cover Images**	
H0	0.2612	0.2405	Pr = 0.5207
H1	0.2387	0.2595	NPV = 0.5208
	Se = 0.5225	Sp = 0.5190	Ex = 0.5208
Kappa = 0.0415			
**10%**	**H0: Stego Images**	**H1: Cover Images**	
H0	0.2504	0.2448	Pr = 0.5057
H1	0.2496	0.2552	NPV = 0.5056
	Se = 0.5008	Sp = 0.5105	Ex = 0.5056
Kappa = 0.0112			
**20%**	**H0: Stego Images**	**H1: Cover Images**	
H0	0.3191	0.1946	Pr = 0.6212
H1	0.1809	0.3054	NPV = 0.6280
	Se = 0.6382	Sp = 0.6108	Ex = 0.6245
Kappa = 0.2490			

**Table 11 entropy-21-00748-t011:** FLD classification evaluation of EEALSBMR algorithm using Moulin features.

**5%**	**H0: Stego Images**	**H1: Cover Images**	
H0	0.2489	0.2476	Pr = 0.5013
H1	0.2511	0.2524	NPV = 0.5012
	Se = 0.4977	Sp = 0.5048	Ex = 0.5012
Kappa = 0.0025			
**10%**	**H0: Stego Images**	**H1: Cover Images**	
H0	0.2559	0.2299	Pr = 0.5268
H1	0.2441	0.2701	NPV = 0.5253
	Se = 0.5117	Sp = 0.5403	Ex = 0.5260
Kappa = 0.0520			
**20%**	**H0: Stego Images**	**H1: Cover Images**	
H0	0.2990	0.1985	Pr = 0.6010
H1	0.2010	0.3015	NPV = 0.6000
	Se = 0.5980	Sp = 0.6030	Ex = 0.6005
Kappa = 0.2010			

**Table 12 entropy-21-00748-t012:** SVM classification evaluation of EEALSBMR algorithm using Farid features.

**5%**	**H0: Stego Images**	**H1: Cover Images**	
H0	0.3438	0.3431	Pr = 0.5005
H1	0.1562	0.1569	NPV = 0.5011
	Se = 0.6876	Sp = 0.3137	Ac = 0.6870
Kappa = 0.0013			
**10%**	**H0: Stego Images**	**H1: Cover Images**	
H0	0.4006	0.3977	Pr = 0.5018
H1	0.0994	0.1023	NPV = 0.5071
	Se = 0.8011	Sp = 0.2046	Ac = 0.5029
Kappa = 0.0057			
**20%**	**H0: Stego Images**	**H1: Cover Images**	
H0	0.3251	0.3199	Pr = 0.5041
H1	0.1749	0.1801	NPV = 0.5074
	Se = 0.6503	Sp = 0.3602	Ac = 0.5052
Kappa = 0.0105			

**Table 13 entropy-21-00748-t013:** SVM classification evaluation of EEALSBMR algorithm using Shi features.

**5%**	**H0: Stego Images**	**H1: Cover Images**	
H0	0.2220	0.2188	Pr = 0.5037
H1	0.2780	0.2812	NPV = 0.5029
	Se = 0.4440	Sp = 0.5625	Ac = 0.5032
Kappa = 0.0065			
**10%**	**H0: Stego Images**	**H1: Cover Images**	
H0	0.2189	0.2161	Pr = 0.5032
H1	0.2811	0.2839	NPV = 0.5024
	Se = 0.4377	Sp = 0.5678	Ac = 0.5028
Kappa = 0.0055			
**20%**	**H0: Stego Images**	**H1: Cover Images**	
H0	0.2282	0.1999	Pr = 0.5330
H1	0.2718	0.3001	NPV = 0.5247
	Se = 0.4564	Sp = 0.6002	Ac = 0.5283
Kappa = 0.0566			

**Table 14 entropy-21-00748-t014:** SVM classification evaluation of EEALSBMR algorithm using Moulin features.

**5%**	**H0: Stego Images**	**H1: Cover Images**	
H0	0.2275	0.2264	Pr = 0.5013
H1	0.2725	0.2736	NPV = 0.5010
	Se = 0.4550	Sp = 0.5472	Ac = 0.5011
Kappa = 0.0023			
**10%**	**H0: Stego Images**	**H1: Cover Images**	
H0	0.2412	0.2380	Pr = 0.5034
H1	0.2588	0.2620	NPV = 0.5031
	Se = 0.4825	Sp = 0.5240	Ac = 0.5032
Kappa = 0.0065			
**20%**	**H0: Stego Images**	**H1: Cover Images**	
H0	0.2922	0.2684	Pr = 0.5212
H1	0.2078	0.2316	NPV = 0.5271
	Se = 0.5844	Sp = 0.4632	Ac = 0.5238
Kappa = 0.0476			

**Table 15 entropy-21-00748-t015:** FLD classification evaluation of EDCT algorithm using Farid features.

**5%**	**H0: Stego Images**	**H1: Cover Images**	
H0	0.2524	0.2454	Pr = 0.5070
H1	0.2476	0.2546	NPV = 0.5069
	Se = 0.5048	Sp = 0.5091	Ac = 0.5070
Kappa = 0.0139			
**10%**	**H0: Stego Images**	**H1: Cover Images**	
H0	0.2617	0.2238	Pr = 0.5390
H1	0.2383	0.2762	NPV = 0.5368
	Se = 0.5234	Sp = 0.5524	Ac = 0.5379
Kappa = 0.0758			
**20%**	**H0: Stego Images**	**H1: Cover Images**	
H0	0.3104	0.1719	Pr = 0.6436
H1	0.1896	0.3281	NPV = 0.6337
	Se = 0.6208	Sp = 0.6562	Ac = 0.6385
Kappa = 0.2770			

**Table 16 entropy-21-00748-t016:** FLD classification evaluation of EDCT algorithm using Shi features.

**5%**	**H0: Stego Images**	**H1: Cover Images**	
H0	0.2548	0.2343	Pr = 0.5209
H1	0.2452	0.2657	NPV = 0.5200
	Se = 0.5095	Sp = 0.5314	Ac = 0.5205
Kappa = 0.0410			
**10%**	**H0: Stego Images**	**H1: Cover Images**	
H0	0.3242	0.1893	Pr = 0.6313
H1	0.1758	0.3107	NPV = 0.6386
	Se = 0.6484	Sp = 0.6213	Ac = 0.6349
Kappa = 0.2697			
**20%**	**H0: Stego Images**	**H1: Cover Images**	
H0	0.4409	0.0635	Pr = 0.8741
H1	0.0591	0.4365	NPV = 0.8807
	Se = 0.8817	Sp = 0.8730	Ac = 0.8773
Kappa = 0.7547			

**Table 17 entropy-21-00748-t017:** FLD classification evaluation of EDCT algorithm using Moulin features.

**5%**	**H0: Stego Images**	**H1: Cover Images**	
H0	0.2611	0.2499	Pr = 0.5110
H1	0.2389	0.2501	NPV = 0.5115
	Se = 0.5223	Sp = 0.5002	Ac = 0.5112
Kappa = 0.0225			
**10%**	**H0: Stego Images**	**H1: Cover Images**	
H0	0.2780	0.2136	Pr = 0.5655
H1	0.2220	0.2864	NPV = 0.5633
	Se = 0.5560	Sp = 0.5728	Ac = 0.5644
Kappa = 0.1288			
**20%**	**H0: Stego Images**	**H1: Cover Images**	
H0	0.3739	0.1243	Pr = 0.7505
H1	0.1261	0.3757	NPV = 0.7487
	Se = 0.7478	Sp = 0.7514	Ac = 0.7496
Kappa = 0.4992			

**Table 18 entropy-21-00748-t018:** SVM classification evaluation of EDCT algorithm using Farid features.

**5%**	**H0: Stego Images**	**H1: Cover Images**	
H0	0.0653	0.0591	Pr = 0.5249
H1	0.4347	0.4409	NPV = 0.5035
	Se = 0.1307	Sp = 0.8817	Ac = 0.5062
Kappa = 0.0124			
**10%**	**H0: Stego Images**	**H1: Cover Images**	
H0	0.0848	0.0644	Pr = 0.5683
H1	0.4152	0.4356	NPV = 0.5120
	Se = 0.1695	Sp = 0.8712	Ac = 0.5204
Kappa = 0.0408			
**20%**	**H0: Stego Images**	**H1: Cover Images**	
H0	0.1734	0.0843	Pr = 0.6729
H1	0.3266	0.4157	NPV = 0.5600
	Se = 0.3469	Sp = 0.8314	Ac = 0.5891
Kappa = 0.1783			

**Table 19 entropy-21-00748-t019:** SVM classification evaluation of EDCT algorithm using Shi features.

**5%**	**H0: Stego Images**	**H1: Cover Images**	
H0	0.3156	0.3138	Pr = 0.5014
H1	0.1844	0.1862	NPV = 0.5024
	Se = 0.6312	Sp = 0.3724	Ac = 0.5018
Kappa = 0.0036			
**10%**	**H0: Stego Images**	**H1: Cover Images**	
H0	0.3572	0.3266	Pr = 0.5224
H1	0.1428	0.1734	NPV = 0.5485
	Se = 0.7145	Sp = 0.3469	Ac = 0.5307
Kappa = 0.0613			
**20%**	**H0: Stego Images**	**H1: Cover Images**	
H0	0.4217	0.2220	Pr = 0.6551
H1	0.0783	0.2780	NPV = 0.7803
	Se = 0.8434	Sp = 0.5560	Ac = 0.6997
Kappa = 0.3994			

**Table 20 entropy-21-00748-t020:** SVM classification evaluation of EDCT algorithm using Moulin features.

**5%**	**H0: Stego Images**	**H1: Cover Images**	
H0	0.3053	0.3020	Pr = 0.5027
H1	0.1947	0.1980	NPV = 0.5042
	Se = 0.6107	Sp = 0.3960	Ac = 0.5033
Kappa = 0.0067			
**10%**	**H0: Stego Images**	**H1: Cover Images**	
H0	0.3021	0.2924	Pr = 0.5082
H1	0.1979	0.2076	NPV = 0.5120
	Se = 0.6042	Sp = 0.4152	Ac = 0.5097
Kappa = 0.0194			
**20%**	**H0: Stego Images**	**H1: Cover Images**	
H0	0.3264	0.2427	Pr = 0.5736
H1	0.1736	0.2573	NPV = 0.5971
	Se = 0.6528	Sp = 0.5147	Ac = 0.5837
Kappa = 0.1674			

**Table 21 entropy-21-00748-t021:** FLD classification evaluation of EDWT algorithm using Farid features.

**5%**	**H0: Stego Images**	**H1: Cover Images**	
H0	0.4786	0.0150	Pr = 0.9695
H1	0.0214	0.4850	NPV = 0.9577
	Se = 0.9571	Sp = 0.9699	Ac = 0.9635
Kappa = 0.9270			
**10%**	**H0: Stego Images**	**H1: Cover Images**	
H0	0.4941	0.0056	Pr = 0.9888
H1	0.0059	0.4944	NPV = 0.9882
	Se = 0.9882	Sp = 0.9888	Ac = 0.9885
Kappa = 0.9770			
**20%**	**H0: Stego Images**	**H1: Cover Images**	
H0	0.4993	0.0005	Pr = 0.9990
H1	0.0007	0.4995	NPV = 0.9987
	Se = 0.9987	Sp = 0.9990	Ac = 0.9989
Kappa = 0.9977			

**Table 22 entropy-21-00748-t022:** FLD classification evaluation of EDWT algorithm using Shi features.

**5%**	**H0: Stego Images**	**H1: Cover Images**	
H0	0.4048	0.0470	Pr = 0.8961
H1	0.0952	0.4530	NPV = 0.8263
	Se = 0.8095	Sp = 0.9061	Ac = 0.8578
Kappa = 0.7156			
**10%**	**H0: Stego Images**	**H1: Cover Images**	
H0	0.4536	0.0311	Pr = 0.9358
H1	0.0464	0.4689	NPV = 0.9100
	Se = 0.9072	Sp = 0.9377	Ac = 0.9225
Kappa = 0.8450			
**20%**	**H0: Stego Images**	**H1: Cover Images**	
H0	0.4753	0.0232	Pr = 0.9534
H1	0.0247	0.4768	NPV = 0.9508
	Se = 0.9507	Sp = 0.9535	Ac = 0.9521
Kappa = 0.9042			

**Table 23 entropy-21-00748-t023:** FLD classification evaluation of EDWT algorithm using Moulin features.

**5%**	**H0: Stego Images**	**H1: Cover Images**	
H0	0.3946	0.0650	Pr = 0.8587
H1	0.1054	0.4350	NPV = 0.8049
	Se = 0.7891	Sp = 0.8701	Ac = 0.8296
Kappa = 0.6592			
**10%**	**H0: Stego Images**	**H1: Cover Images**	
H0	0.4394	0.0387	Pr = 0.9191
H1	0.0606	0.4613	NPV = 0.8839
	Se = 0.8789	Sp = 0.9227	Ac = 0.9008
Kappa = 0.8015			
**20%**	**H0: Stego Images**	**H1: Cover Images**	
H0	0.4603	0.0321	Pr = 0.9348
H1	0.0397	0.4679	NPV = 0.9218
	Se = 0.9206	Sp = 0.9358	Ac = 0.9282
Kappa = 0.8564			

**Table 24 entropy-21-00748-t024:** SVM classification evaluation of EDWT algorithm using Farid features.

**5%**	**H0: Stego Images**	**H1: Cover Images**	
H0	0.4770	0.0230	Pr = 0.9541
H1	0.0230	0.4770	NPV = 0.9541
	Se = 0.9541	Sp = 0.9541	Ac = 0.9541
Kappa = 0.9082			
**10%**	**H0: Stego Images**	**H1: Cover Images**	
H0	0.4893	0.0058	Pr = 0.9883
H1	0.0107	0.4942	NPV = 0.9789
	Se = 0.9787	Sp = 0.9884	Ac = 0.9835
Kappa = 0.9670			
**20%**	**H0: Stego Images**	**H1: Cover Images**	
H0	0.4984	0.0084	Pr = 0.9835
H1	0.0016	0.4916	NPV = 0.9967
	Se = 0.9968	Sp = 0.9832	Ac = 0.9900
Kappa = 0.9800			

**Table 25 entropy-21-00748-t025:** SVM classification evaluation of EDWT algorithm using Shi features.

**5%**	**H0: Stego Images**	**H1: Cover Images**	
H0	0.3366	0.1658	Pr = 0.6700
H1	0.1634	0.3342	NPV = 0.6716
	Se = 0.6731	Sp = 0.6684	Ac = 0.6708
Kappa = 0.3415			
**10%**	**H0: Stego Images**	**H1: Cover Images**	
H0	0.4107	0.1371	Pr = 0.7497
H1	0.0893	0.3629	NPV = 0.8024
	Se = 0.8213	Sp = 0.7257	Ac = 0.7735
Kappa = 0.5470			
**20%**	**H0: Stego Images**	**H1: Cover Images**	
H0	0.4605	0.1175	Pr = 0.7967
H1	0.0395	0.3825	NPV = 0.9063
	Se = 0.9210	Sp = 0.7650	Ac = 0.8430
Kappa = 0.6859			

**Table 26 entropy-21-00748-t026:** SVM classification evaluation of EDWT algorithm using Moulin features.

**5%**	**H0: Stego Images**	**H1: Cover Images**	
H0	0.3707	0.1108	Pr = 0.7699
H1	0.1293	0.3892	NPV = 0.7506
	Se = 0.7413	Sp = 0.7785	Ac = 0.7599
Kappa = 0.5198			
**10%**	**H0: Stego Images**	**H1: Cover Images**	
H0	0.4332	0.0725	Pr = 0.8567
H1	0.0668	0.4275	NPV = 0.8649
	Se = 0.8665	Sp = 0.8550	Ac = 0.8608
Kappa = 0.7215			
**20%**	**H0: Stego Images**	**H1: Cover Images**	
H0	0.4672	0.0724	Pr = 0.8659
H1	0.0668	0.4276	NPV = 0.9288
	Se = 0.9345	Sp = 0.8552	Ac = 0.8949
Kappa = 0.7897			

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
