# Peer review of "Comparative Study of Three Steganographic Methods Using a Chaotic System and Their Universal Steganalysis Based on Three Feature Vectors"

_entropy, 2019, doi:10.3390/e21080748_

Round 1

Reviewer 1 Report

In this manuscript the authors are proposing improved realization of three steganographic methods. The base methods are well described and the security enhancements are well explained. The introduction provides the necessary theoretical background concerning the steganography. The second section describes the chaotic system using Cat map. The third section provides the enhanced steganographic algorithms - Enhanced EALSBMR, Enhanced DCT and Enhanced DWT. The next three section of the paper provide extensive steganographic analysis.

As general the study in this paper is innovative with the necessary scientific soundness and interesting to readers and researchers. My notes concerning the manuscript are as follows:

1.      In section two: Description of the proposed chaotic system – chaotic generators are always provided with test for randomness to prove the provided pseudorandom sequence is indeed random. If those test are made in different research, the authors should provide reference with the tests.

2.      The provided steganographic analysis is very extensive, but the results are not compared with similar research. For examples PSNR can be compared with:

Kordov, K., Stoyanov, B. (2017). Least Significant Bit Steganography using Hitzl-Zele Chaotic Map. International Journal of Electronics and Telecommunications, 63(4), 417-422.

Stoyanov, B. P., Zhelezov, S. K., Kordov, K. M. (2016). Least significant bit image steganography algorithm based on chaotic rotation equations. Comptes rendus de l’Academie bulgare des Sciences, 69(7), 845-850.

Chan, C.K.; Cheng, L. Hiding data in images by simple LSB substitution. Pattern Recognition 2004,

37, 469–474

Wu, H.C.; Wu, N.I.; Tsai, C.S.; Hwang, M.S. Image steganographic scheme based on pixel-value differencing and LSB replacement methods. IEE Proceedings - Vision, Image, and Signal Processing 2005, 152, 611.

Stego capacity can be compared with

Wu, H.C.; Wu, N.I.; Tsai, C.S.; Hwang, M.S. Image steganographic scheme based on pixel-value differencing and LSB replacement methods. IEE Proceedings - Vision, Image, and Signal Processing 2005, 152, 611.

Jung, K.; Ha, K.; Yoo, K. Image Data Hiding Method Based on Multi-Pixel Differencing and LSB Substitution Methods. 2008 International Conference on Convergence and Hybrid Information Technology, 2008, pp. 355–358. doi:10.1109/ICHIT.2008.279

SSIM also can be compared with some of the references in the paper.

Author Response

Dear Reviewer,

Thank you so much for your kind comments to our work.

Could you please find the answers to your comments in the attached file.

Best regards

Safwan El Assad

Reviewer 2 Report

In this manuscript, the security of three steganographic methods is enhanced by using a chaotic system. The first steganographic method is in the spatial domain, while the other two methods are in the frequency domain (i.e., DCT and DWT). The adopted chaotic system is robust and used to secure the hidden message content in case of its detection. Three blind steganalysis methods, based on multi-resolution wavelet decomposition, are used to distinguish the stego images from the cover ons. The detailed detections results are provided to show the performances of three steganographic methods compared.

As a comparative study on three steganographic methods using a chaotic system has been conducted with steganalysis results, the manuscript serves as a good tutorial on steganography and steganalysis.

Secondly, the aspect of security enhancement should be focused in the manuscript with the compared evaluation results.

Thirdly, analyze why the EDWT steganographic method can be detected more easily (partly due to that the DWT-compressed images are not good covers for steganography?).

Author Response

(The authors gave the same response as above.)

Round 2

Reviewer 1 Report

The authors have improved their manuscript and the reviewers’ comments are taken under consideration.